# Modulation of let-7 miRNAs controls the differentiation of effector CD8 T cells

Alexandria C Wells[1], Keith A Daniels[2], Constance C Angelou[1], Eric Fagerberg[1], Amy S Burnside[1], Michele Markstein[3], Dominique Alfandari[1], Raymond M Welsh[2], Elena L Pobezinskaya[1]*, Leonid A Pobezinsky[1]*

[1]Department of Veterinary and Animal Sciences, University of Massachusetts, Amherst, United States; [2]Department of Pathology, University of Massachusetts Medical School, Worcester, United States; [3]Department of Biology, University of Massachusetts, Amherst, United States

**Abstract** The differentiation of naive CD8 T cells into effector cytotoxic T lymphocytes upon antigen stimulation is necessary for successful antiviral, and antitumor immune responses. Here, using a mouse model, we describe a dual role for the let-7 microRNAs in the regulation of CD8 T cell responses, where maintenance of the naive phenotype in CD8 T cells requires high levels of let-7 expression, while generation of cytotoxic T lymphocytes depends upon T cell receptor-mediated let-7 downregulation. Decrease of let-7 expression in activated T cells enhances clonal expansion and the acquisition of effector function through derepression of the let-7 targets, including Myc and Eomesodermin. Ultimately, we have identified a novel let-7-mediated mechanism, which acts as a molecular brake controlling the magnitude of CD8 T cell responses.

*For correspondence:
pobezinskaya@umass.edu (ELP);
lpobezinsky@umass.edu (LAP)

**Competing interests:** The authors declare that no competing interests exist.

## Introduction

CD8 T cells are responsible for the rapid clearance of virally infected, and cancerous cells in the organism. Prior to encounter with antigen, naive CD8 T cells exhibit a quiescent state that is characterized by very low rates of proliferation, and a 'quiet' transcriptional landscape with no expression of any effector molecules. After antigen recognition, activated CD8 T cells undergo blastogenesis and rapid clonal expansion, which is followed by differentiation into effector cytotoxic T lymphocytes (CTLs), and memory T cells that are both capable of producing effector cytokines and killing target cells. These changes in T cells are accompanied by metabolic reprograming from oxidative phosphorylation to aerobic glycolysis that provides energy and larger amounts of the biomacromolecular intermediates needed to support growth (*Ward and Thompson, 2012*).

The productive differentiation of CD8 T cells requires three definitive signals; one is from the T cell receptor (TCR) during antigen recognition, the second is from co-stimulation, and the third is due to cytokines. It has been extensively studied how these signals execute the CD8 T cell differentiation program by activation of a complex network of transcription factors such as Myc, Notch-1, T-bet, Eomes, Blimp-1, Zbtb32, Runx3, and Zeb2 (*Backer et al., 2014*; *Best et al., 2013*; *Pearce et al., 2003*; *Szabo et al., 2000*). However, the global regulation of these processes is not yet fully understood.

Post-transcriptional regulatory mechanisms provide broad regulation of gene expression. The most efficient post-transcriptional regulatory machinery involves RNA interference mediated by microRNAs (miRNAs), small non-coding RNAs that target mRNA in a sequence-specific manner to prevent protein synthesis, and the expression of which is both inducible, and tissue- specific (*Bartel, 2009*; *Ha and Kim, 2014*). The importance of this type of regulation for T cells was first demonstrated through the ectopic expression of various miRNAs during hematopoiesis (*Chen et al., 2004*).

Later, it was found that miRNA depletion achieved by a T-cell-specific deletion of Dicer, a critical enzyme involved in the biogenesis of mature miRNAs, resulted in severe defects, including aberrant proliferation, differentiation, and function of T lymphocytes (*Cobb et al., 2006*, *Cobb et al., 2005*; *Muljo et al., 2005*). Further research corroborated these earlier studies, identifying the role of specific miRNAs in these processes. MiR-181 was shown to modulate TCR sensitivity and signal strength in developing and mature T cells (*Henao-Mejia et al., 2013*; *Li et al., 2007*). In CD4 T cells, miR-29 was shown to control the T helper response through the direct targeting of IFN-γ mRNA (*Ma et al., 2011*), as well as the transcription factors T-bet and Eomes (*Steiner et al., 2011*). In CD8 T cells, the upregulation of miR-130/301 (*Zhang and Bevan, 2010*) and the miR-17–92 cluster (*Steiner et al., 2011*; *Katz et al., 2014*; *Wu et al., 2012*) was demonstrated to be essential for the initiation of differentiation upon antigen stimulation.

Let-7 miRNAs are the largest and most abundant family of miRNAs in lymphocytes, yet very little is know about their function. It has been shown that let-7 miRNAs are needed for the differentiation and function of natural killer T cells, an innate-like subset of T cells (*Pobezinsky et al., 2015*; *Yuan et al., 2012*). It has also been suggested that let-7 miRNAs may regulate T helper responses (*Polikepahad et al., 2010*; *Swaminathan et al., 2012*) and play a role in the suppressive function of Tregs (*Okoye et al., 2014*). However, the extent to which let-7 miRNAs regulate the differentiation or function of CD8 T cells has yet to be explored.

In the present study, we examine the role of let-7 miRNA expression in naive and differentiating CD8 T cells. We found that in CD8 T cells high levels of let-7 miRNAs are necessary to maintain the naive phenotype, while TCR-mediated down-regulation of let-7 levels in activated cells is critical for the effective differentiation and function of CTLs. In fact, experimentally forced let-7 expression severely impairs the proliferation and differentiation of CD8 T cells, while let-7 deficiency significantly enhances the cytotoxic function of CTLs, and consequently immune responses in vivo. Given these findings, we propose a model in which let-7 acts as a molecular hub by converting the strength of TCR signaling into the strength of CD8 T cell function.

## Results

### let-7 miRNA expression maintains the quiescent state in naive CD8 T cells

To explore the potential regulatory role of let-7 miRNAs in CD8 T cells, the expression levels of let-7 miRNA family members in naive and activated CD8 T cells were determined and normalized to two different housekeeping RNAs. Surprisingly, the initially very high expression of let-7 miRNAs in naive CD8 T cells was reduced by TCR signaling, and this downregulation was proportional to the strength and duration of TCR-stimulation, regardless of housekeeping RNA used (*Figure 1A,B* and *Figure 1— figure supplement 1* and *Figure 1—figure supplement 2A*). As a specificity control, we confirmed the upregulation of miR-17 from the miR-17–92 locus (*Figure 1—figure supplement 2B*) that is induced upon T cell activation (*Katz et al., 2014*; *Wu et al., 2012*). Taken together, these results suggest that TCR-mediated signaling inhibits the expression of let-7 miRNAs during T cell activation.

To determine the functional significance of let-7 expression in naive CD8 lymphocytes, we examined CD8 T cells from P14[+]Lin28Tg*Rag2*[−/−] mice where T-cell-specific expression of the Lin28 protein blocks let-7 biogenesis (*Pobezinsky et al., 2015*; *Piskounova et al., 2011*), inhibiting let-7 expression, and P14 is a monoclonal T cell receptor specific to the lymphocytic choriomeningitis virus (LCMV) peptide gp33-41, presented in the context of H-2D[b] molecules (*Figure 1—figure supplement 3A*). In comparison to the P14[+]wild-type counterparts, P14[+]Lin28Tg CD8 T cells were significantly larger in size, with a dramatically increased proportion of Ki67-positive cells (*Cuylen et al., 2016*) (*Figure 1C,D*). Consistent with the increased proportion of Ki67-positive cells, P14[+]Lin28Tg mice had a significantly higher frequency of BrdU-positive cells in both the spleen and lymph nodes, as compared to P14[+]wild-type mice (*Figure 1E*). In addition, surface expression of T cell activation markers, such as the IL-2 receptor beta-chain (CD122), and CD44 was also increased, while cells remained CD25 negative (*Figure 1—figure supplement 3B*). Thus, these results suggest that the expression of let-7 miRNAs may maintain the quiescent state in naive CD8 T cells.

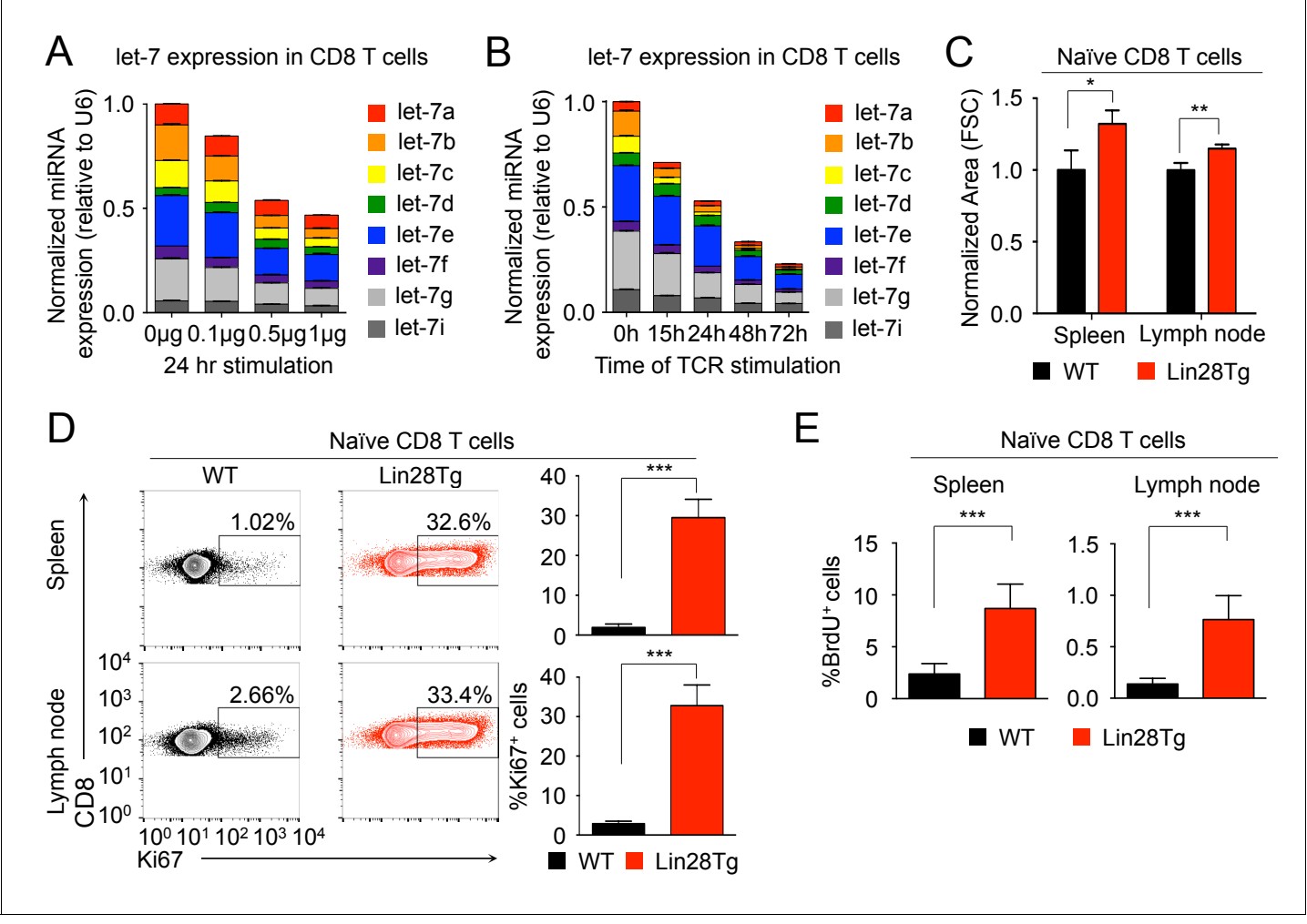

**Figure 1.** let-7 expression is necessary and sufficient to maintain the naive phenotype of CD8 T cells prior to TCR stimulation. (A) Quantitative RT-PCR analysis of individual let-7 miRNAs in naive CD8 T cells stimulated with plate-bound anti-TCR (µg as indicated) and anti-CD28 (5 µg) for 24 hr, presented relative to results obtained for the small nuclear RNA (U6 control) and normalized to the unstimulated (0 µg) sample. (B) Quantitative RT-PCR analysis of individual let-7 miRNAs in naive CD8 T cells stimulated with plate-bound anti-TCR (5 µg) and anti-CD28 (5 µg) over increasing periods of time as indicated, presented relative to results obtained for the small nuclear RNA (U6 control) and normalized to the unstimulated (0 hr) sample. (C) Size analysis based on FSC (forward scatter) of naive CD8 T cells from the spleens and lymph nodes of P14+wild-type and P14+Lin28Tg mice, both on $Rag2^{-/-}$ background, normalized to wild-type. (D) Expression of Ki67 in naive CD8 T cells from spleens and lymph nodes of P14+wild-type and P14+Lin28Tg mice, both on $Rag2^{-/-}$ background (left). Quantification of the frequency of Ki67+ cells in these populations (right). (E) Frequency of BrdU+ cells in naive CD8 T cells from spleens and lymph nodes of P14+wild-type (n = 6) and P14+Lin28Tg (n = 5) mice, both on $Rag2^{-/-}$ background, labeled with BrdU in vivo. *p<0.05, **p<0.01,***p<0.001, compared with wild-type using two-tailed Student's *t*-test. Data are one experiment representative of three independent experiments (a, b; mean and s.e.m. of technical triplicates; c, d; mean and s.e.m. of three experiments).

The following source data and figure supplements are available for figure 1:

**Source data 1.** Quantification of the expression of miRNAs by qPCR, and flow cytometry analysis of the phenotype of naive CD8 T cells.

**Figure supplement 1.** let-7 expression is downregulated upon TCR stimulation in CD8 T cells.

**Figure supplement 2.** TCR-mediated regulation of miRNA expression.

**Figure supplement 3.** Expression of let-7 and activation markers in P14+ CD8 T cells.

## let-7 miRNA expression in CTLs affects both the antiviral and antitumor immune responses

TCR stimulation of naive T cells leads to a rapid loss of the quiescent state and differentiation into effector cells. Given that the expression of let-7 miRNAs, which is critical for the maintenance of naive CD8 T cells, is inhibited by TCR signaling (*Figure 1A,B*), we hypothesized that the downregulation of let-7 miRNAs in response to TCR stimulation is necessary for the differentiation and function of effector CD8 T cells (*Figure 2A*).

To determine whether TCR-mediated downregulation of let-7 miRNAs is required for CD8 T cell differentiation in vivo, we analyzed the fate of P14$^+$ CD8 T cells with forced let-7 expression in response to acute viral infection with LCMV Armstrong. The doxycycline-inducible let-7g transgene (*Zhu et al., 2011*) maintains let-7g miRNA expression in lymphocytes in the presence of doxycycline, even after TCR stimulation (*Figure 2—figure supplement 1A*). Donor CD45.2$^+$CD8$^+$ T cells from P14$^+$ and P14$^+$ let-7 transgenic (let-7Tg) mice were adoptively transferred into host congenic wild-type CD45.1$^+$ mice that were concurrently infected with LCMV, and the differentiation state of P14$^+$ cells was assessed 7 days post- injection. Interestingly, the recovery of donor CD8 T cells at the peak of the immune response revealed that P14$^+$let-7Tg CD8 T cells failed to clonally expand (*Figure 2B*) and lacked KLRG1 expression, an established marker of terminal effector CTLs (*Dominguez et al., 2015*; *Joshi et al., 2007*; *Thimme et al., 2005*; *Voehringer et al., 2001*) (*Figure 2C*). Furthermore, let-7Tg CTLs had a reduced frequency of IFN-γ$^+$TNF-α$^+$ cytokine double-producing cells, a hallmark of an effective CD8 T cell response (*Kaech et al., 2002*; *Wherry and Ahmed, 2004*; *Williams and Bevan, 2007*), while the differentiation of endogenous host-derived CTLs was normal, suggesting a cell-intrinsic mechanism (*Figure 2C*). Importantly, mRNAs of the *Klrg1*, *Ifng* and *Tnfa* genes are not targets of let-7 miRNAs, therefore the reduced frequencies of effector cells generated from let-7Tg CD8 T cells is not simply a result of direct suppression of effector molecule expression. Thus, sustained let-7 expression following TCR activation severely impaired the clonal expansion and differentiation of CTLs in response to viral infection in vivo.

As the allogeneic response to foreign MHC is one of the most robust responses of the immune system, we further aimed to determine whether steady levels of let-7 in T cells would suppress the allogeneic response in vivo (*Felix and Allen, 2007*; *Janković et al., 2002*). We used the P815 mastocytoma, an allogeneic (H-2d haplotype) tumor, that has been shown to elicit a CD8 T-cell-mediated allogeneic immune response (*Zhan et al., 2000*). We confirmed this by demonstrating that P14$^+$*Rag2*$^{-/-}$ mice failed to reject the tumor without adoptive transfer of CD8 T cells (*Figure 2—figure supplement 1B*). Interestingly, when the P815 cells were injected into wild-type or let-7Tg mice of H-2b haplotype, 60% of let-7Tg mice were unable to effectively respond to, and clear the tumor (*Figure 2D*). Additionally, at 7 days post-injection, wild-type mice retained on average 40 × 10$^6$ cancer cells in the peritoneal cavity, while let-7Tg mice contained 115 × 10$^6$ cancer cells (*Figure 2E*). We concluded that CD8 T cells that maintained let-7 expression upon stimulation and differentiation had a compromised response to the alloantigen, and thus failed to reject P815 tumor cells. Taken together, our results demonstrate that the decrease in let-7 expression upon TCR activation is necessary for the proper proliferation and differentiation of cytotoxic CD8 T cells in vivo.

## Expression of let-7 miRNAs in activated CD8 T cells inhibits proliferation and the gene expression program responsible for the metabolic switch

To elucidate the underlying mechanisms of let-7-mediated suppression of CD8 T-cell responses (*Figure 2*), we first analyzed the impact of let-7 miRNAs on T cell clonal expansion. Sorted naive (CD44$^{lo}$CD25$^-$) CD8 T cells with different levels of let-7 expression (*Figure 3—figure supplement 1A,B*) were activated with anti-CD3 mAbs in vitro for 3 days. Interestingly, let-7Tg CD8 T cells proliferated less than their wild-type counterparts, while Lin28Tg CD8 T cells exhibited enhanced proliferation (*Figure 3A*). These results indicate that let-7 miRNAs negatively regulate clonal expansion of activated CD8 T cells, which is consistent with our previous results in vivo (*Figure 2B*). Thus, we concluded that TCR-mediated downregulation of let-7 expression is needed for successful proliferation of activated CD8 T cells.

Let-7 miRNAs are well-documented tumor suppressors. It has been shown that let-7 inhibits proliferation in cancerous cells by directly targeting the mRNA of genes that are involved in the

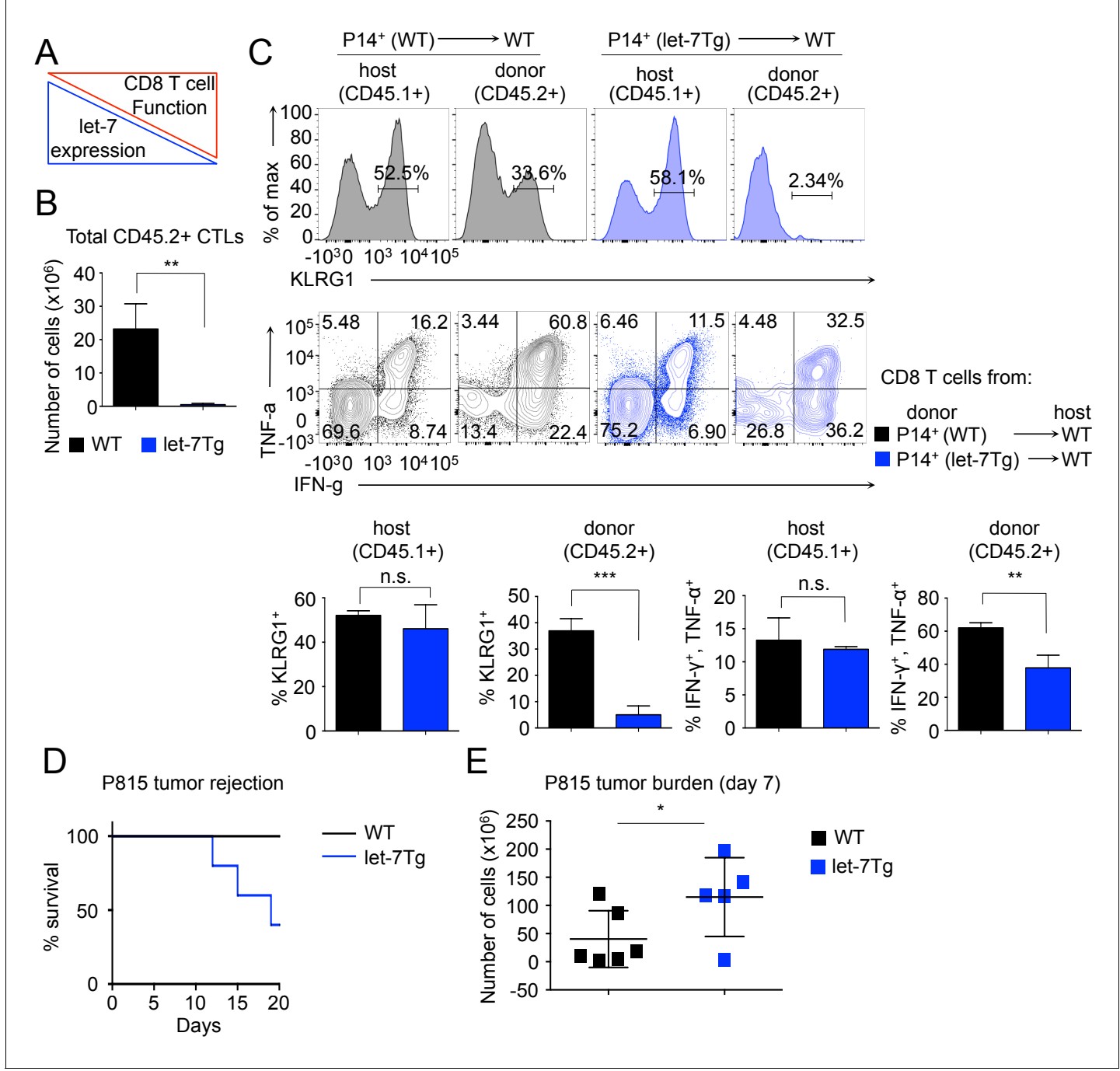

**Figure 2.** Inhibitory role of let-7 miRNA expression in CD8 T-cell-mediated responses in vivo. (**A**) Schematic representation of the hypothesis that let-7 expression inhibits the differentiation of CD8 T cells. (**B**) Quantification of the number of donor (CD45.2[+]) CD8 T cells from P14[+]wild-type (n = 3) or P14[+]let-7Tg mice (n = 3) in the spleens of congenic (CD45.1[+]) host mice 7 days after cell transfer and LCMV Armstrong infection. (**C**) Surface expression of the activation marker KLRG1 on wild-type host, and indicated donor cells (top). Expression of IFN-γ, and TNF-α in wild-type host, and donor LCMV-specific CD8 T cells from the indicated mice, as determined by re-stimulation with the gp33 peptide, and subsequent intracellular staining (middle). Quantification of the frequency of KLRG1[+], and IFN-γ[+], TNF-α[+] populations in wild-type host, and donor cells from the indicated mice (bottom). (**D**) Analysis of the survival of wild-type (n = 5) or let-7Tg (n = 5) mice injected i.p. with 30 × 10[6] P815 cells. (**E**) Quantification of the number of P815 tumor cells remaining in the peritoneal cavity 7 days after i.p. injection of 20 × 10[6] cells into either wild-type (n = 6), or let-7Tg mice (n = 5). n.s., not significant (p>0.05), *p<0.05, **p<0.01, and ***p<0.001, compared with wild-type using two-tailed Student's t-test (**B, C, D**) or one-tailed Student's t-test (**E**). Data are from one experiment representative of three independent experiments (b, c; mean and s.e.m. of technical triplicates), or two experiments (**D,E**).

*Figure 2 continued on next page*

*Figure 2 continued*

The following source data and figure supplement are available for figure 2:

**Source data 1.** Quantification of let-7 expression by qPCR, and flow cytometry analysis of effector CD8 T cells during the response to LCMV infection.
**Figure supplement 1.** P815 (H-2d) tumor rejection is mediated by CD8 T cells in H-2b mice.

regulation of the cell cycle (*Johnson et al., 2007*). In fact, the expression of some let-7 targets such as phosphatase cdc25a (*cdc25a*), kinase cdk6 (*cdk6*), and cyclin D2 (*ccnd2*), was suppressed in activated let-7Tg CD8 T cells, as compared to Lin28Tg CD8 T cells where it was derepressed (*Figure 3B*). It has been shown that let-7 may also regulate the transcription factor Myc, expression of which is upregulated upon T cell activation and is essential for CD8 T cell proliferation (*Best et al., 2013*; *Iritani et al., 2002*; *Kim et al., 2009*; *Nie et al., 2012*; *Verbist et al., 2016*). Interestingly, activated let-7Tg CD8 T cells had reduced expression of Myc, whereas let-7-deficient (Lin28Tg) CD8 T cells had increased Myc expression (*Figure 3C*). To demonstrate that Myc activity was suppressed by let-7, we also analyzed the expression of a direct transcriptional target of Myc, the transcription factor AP4 (*Tfap4*), which sustains Myc-mediated effects in CD8 T cells during the later stages of differentiation (*Chou et al., 2014*). Although mRNA of the *Tfap4* gene is not a target of let-7, the expression of *Tfap4* mRNA was significantly reduced in let-7Tg CD8 T cells, and enhanced in Lin28Tg CD8 T cells (*Figure 3C*), suggesting that let-7 regulates Myc activity in CD8 T cells.

Another important function of Myc in activated CD8 T cells is to support the proliferative burst through the metabolic reprogramming of lymphocytes from primarily oxidative phosphorylation (resting) to glycolysis (activated), as well as through an increase in protein synthesis (*Cham et al., 2008*; *Wang et al., 2011*). To test whether let-7 miRNAs have an impact on the metabolic switch in activated CD8 T cells through its regulation of Myc, we assessed the expression of key glucose transporters, glycolytic enzymes, and protein synthesis enzymes that have been established as direct targets of Myc in activated CD8 T cells (*Wang et al., 2011*). Strikingly, the expression of all tested targets was suppressed in let-7 transgenic CD8 T cells, and increased in Lin28Tg-activated lymphocytes, suggesting that let-7 expression may influence Myc-dependent metabolic reprogramming of activated CD8 T cells (*Figure 3D*). Taken together, these results demonstrate that let-7 miRNAs control the proliferation of activated CD8 T cells by modulating the expression and activity of genes that are involved in the regulation of the cell cycle and metabolism.

## let-7 expression regulates the function of differentiated CD8 T cells

To identify whether let-7-mediated suppression of CD8 T cell immune responses is due to a failure to acquire effector function in addition to a proliferative defect, we assessed the cytotoxic activity of CTLs generated from P14+wild-type, P14+let-7Tg, and P14+Lin28Tg mice. In fact, the expression of the let-7 transgene in P14+ CTLs greatly diminished their cytotoxic activity (*Figure 4A*). Alternatively, P14+Lin28Tg CTLs exhibited enhanced cytotoxicity (*Figure 4A*), which could be reduced by restoring let-7 expression through the induction of the doxycycline-inducible let-7 transgene in P14+let-7TgLin28Tg (4 Tg) CTLs (*Figure 4B*), demonstrating that let-7 deficiency, and not Lin28 overexpression, is responsible for increased cytotoxicity. Thus, we demonstrated that TCR-mediated downregulation of let-7 microRNAs is necessary for the acquisition of cytotoxic function in differentiating CD8 T cells.

To investigate the mechanism of how let-7 miRNA expression impacts CD8 T cell function, we examined the phenotype of in vitro generated effector CTLs from wild-type, let-7Tg, and Lin28Tg mice. Let-7Tg CTLs had less internal complexity based on the intensity of side scatter (SSC) than wild-type cells, whereas Lin28Tg CD8 effector cells had significantly greater complexity (*Figure 4C*), suggesting a change in the number of cytotoxic granules. Indeed, let-7Tg CTLs contained fewer Granzyme A and Granzyme B positive granules than wild-type CTLs, while Lin28Tg CTLs had more (*Figure 4D*). These results indicated that the expression of let-7 controls the quantity of cytotoxic granules produced during the differentiation of CTLs.

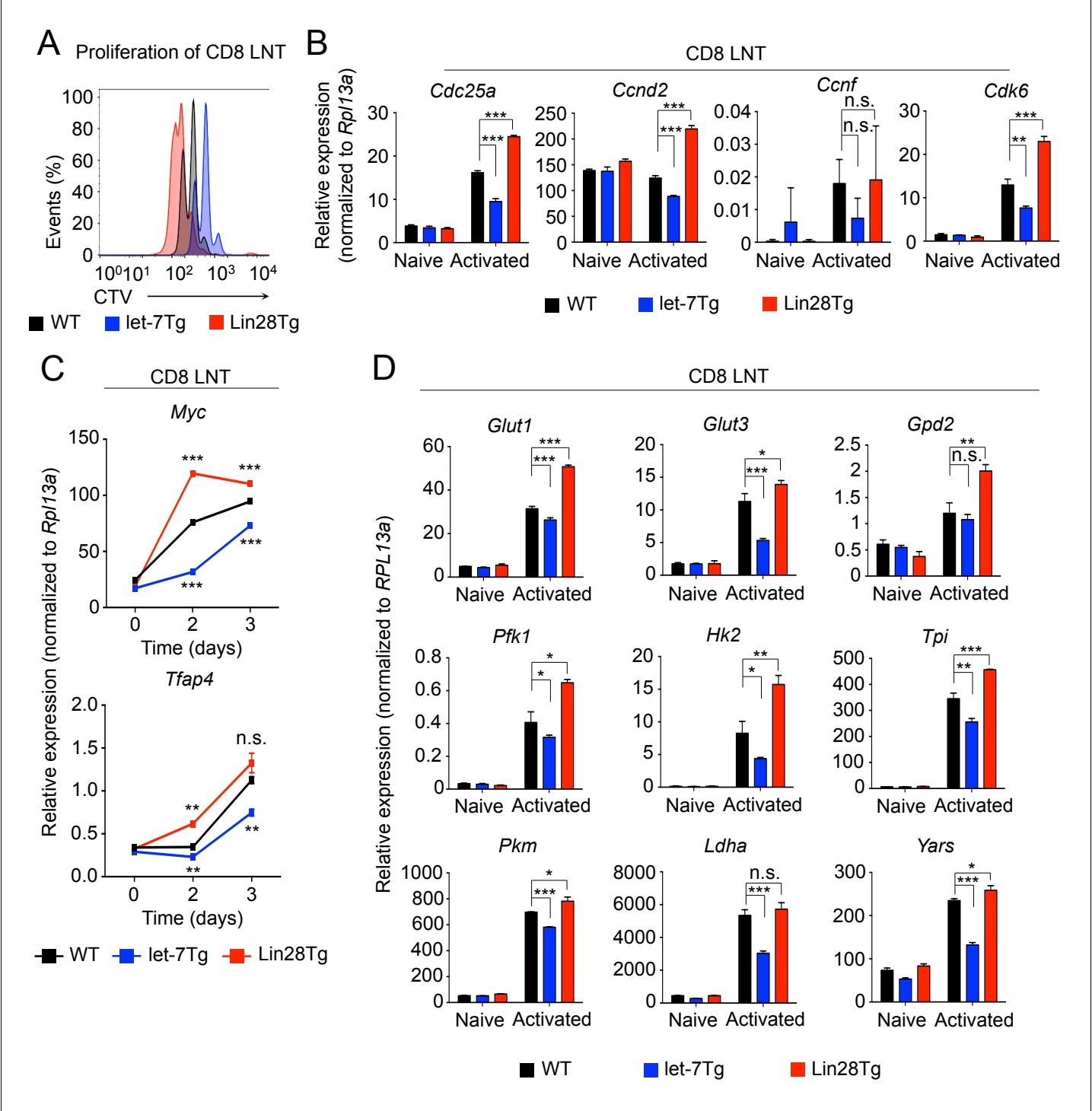

**Figure 3.** let-7 miRNAs suppress the proliferation and metabolism of activated CD8 T cells. (**A**) Analysis of the proliferation of CellTrace Violet-labeled naive CD8 T cells from the indicated mice 72 hr after activation with anti-CD3 mAbs. (**B**) Quantitative RT-PCR analysis of cell cycle regulators: *Cdc25a* (Cell division cycle 25A phosphatase), *Ccnd2* (Cyclin D2), *Ccnf* (Cyclin F), *Cdk6* (Cyclin dependent kinase 6) in naive and activated wild-type, let-7Tg, and Lin28Tg CD8 T cells 3 days after anti-CD3 mAb stimulation, presented relative to the ribosomal protein *Rpl13a*. (**C**) Quantitative RT-PCR analysis of *Myc* and *Tfap4* (Transcription factor AP-4) in CD8 T cells after stimulation with anti-CD3 mAbs, presented relative to the ribosomal protein *Rpl13a*. Wild-type, let-7Tg, and Lin28Tg CD8 T cells were stimulated with anti-CD3 mAbs and differentiated for the indicated time. (**D**) Quantitative RT-PCR analysis of the expression of genes involved in the metabolic switch: *Glut1* (Glucose transporter 1), *Glut3* (Glucose transporter 3), *Gpd2* (Glycerol-3-phosphate dehydrogenase 2), *Pfk1* (Phosphofructokinase 1), *Hk2* (Hexokinase 2), *Tpi* (Triose phosphate isomerase), *Pkm* (Pyruvate kinase muscle isozyme), *Ldha* (Lactate dehydrogenase A), *Yars* (Tyrosyl-tRNA synthetase) in wild-type, let-7Tg, and Lin28Tg CD8 T cells 3 days after stimulation with anti-CD3 mAbs,

*Figure 3 continued on next page*

Figure 3 continued

presented relative to the ribosomal protein, *Rpl13a*. n.s., not significant (p>0.05), *p<0.05, **p<0.01, and ***p<0.001, compared with wild-type using two-tailed Student's *t*-test. (a, b, c, d; one experiment representative of three independent experiments (a) or mean and s.e.m. of technical triplicates (b, c, d)).

The following source data and figure supplement are available for figure 3:

**Source data 1.** Quantification of the expression of cell cycle regulators, *Myc* and *Tfap4*, metabolic enzymes, and let-7 miRNAs by qPCR.

**Figure supplement 1.** Sorting strategy and let-7 expression in polyclonal CD8 T cells.

Next, to determine whether let-7 expression in CTLs influences the number of granules by controlling the expression of effector molecules, gene expression of Granzyme A (*Gzma*), Granzyme B (*Gzmb*), and Perforin (*Prf1*), the key cytolytic factors in cytotoxic granules (*Hayes et al., 1989*; *Lancki et al., 1991*), was measured. Let-7Tg CTLs, which contained fewer cytotoxic granules, expressed less mRNA for *Gzma*, *Gzmb*, and *Prf1*, as compared to wild-type cells, while Lin28Tg CTLs had higher expression of these effector molecules (*Figure 4E*). Moreover, the observed changes in mRNA expression of effector molecules, including Interferon-gamma (IFN-γ), were consistent at the protein level (*Figure 4F,G*). Importantly, the induction of let-7 expression in the presence of Lin28 in P14+let-7TgLin28Tg (4 Tg) CTLs reduced the expression of these effector molecules (*Figure 4—figure supplement 1*), again demonstrating that enhanced effector molecule expression is due to let-7 deficiency. Together, these data indicate that let-7 miRNAs negatively regulate CTL function by preventing the expression of important cytotoxic molecules.

## let-7 miRNAs directly target the 'master regulator' transcription factor, Eomesodermin during CTL differentiation

To determine how let-7 regulates the differentiation and function of CTLs, we considered the possibility that let-7 miRNAs may directly regulate the expression of effector molecules. Although typical miRNA-binding sites are found in the 3' untranslated regions (UTRs) of mRNA transcripts (*Lai, 2002*; *Reinhart et al., 2000*; *Wightman et al., 1993*), we analyzed the full length mRNA of *Prf1*, *Gzma*, *Gzmb*, and *Ifng*, yet no let-7-binding sites were found. This suggested that let-7 may indirectly regulate the expression of these molecules by controlling more global regulators, such as transcription factors. It is known that the expression of effector molecules and cytotoxic function of CD8 T cells are tightly regulated by a group of transcription factors, including Eomesodermin (Eomes), T-bet, Zbtb32, Runx3, Notch-1 and Blimp-1 (*Backer et al., 2014*; *Pearce et al., 2003*; *Cho et al., 2009*; *Kallies et al., 2009*; *Rutishauser et al., 2009*; *Szabo et al., 2002*; *Shin et al., 2014*). To determine if these factors are regulated by let-7, relative gene expression in CTLs with different levels of let-7 was assessed for *Eomes*, *Tbx21*, *Zbtb32*, *Prdm1*, *Runx3d* and Notch-1 genes. The only transcription factor whose expression on both the mRNA and protein levels was reduced in let-7Tg CTLs, and enhanced in Lin28Tg CTLs, was *Eomes* (*Figure 5A,B* and *Figure 5—figure supplement 1A,B*). Furthermore, the induction of let-7 expression in the presence of Lin28 in P14+let-7TgLin28Tg (4 Tg) CTLs reduced the expression of *Eomes* (*Figure 5C*), demonstrating that the Lin28-mediated knockdown of let-7 miRNAs results in increased expression of *Eomes* in Lin28Tg CTLs. Of note, we also found that T-bet expression was inversely correlated with Eomes, suggesting that an Eomes-dependent mechanism may control T-bet expression (*Figure 5A,B*). Thus, these results demonstrate that let-7 miRNAs negatively regulate *Eomes* expression.

Next, we wanted to determine whether let-7 miRNAs can target *Eomes* mRNA. Interestingly, there is no let-7-binding site located in the 3'UTR of *Eomes*, but rather a conserved binding motif within the open reading frame of *Eomes* mRNA was identified (*Figure 5D*; *Supplementary file 1*). To determine whether this binding site is functional, the 10-nucleotide mouse *Eomes*-let-7-binding motif was cloned into a dual luciferase vector. The vector was then transfected into NIH 3T3 fibroblasts, which have high endogenous expression of the let-7 family members (*Figure 5—figure supplement 2A,B*). This resulted in a significant decrease in luciferase activity, indicating that let-7 can directly bind *Eomes* (*Figure 5E*). When this sequence was mutated, disrupting the let-7-binding site,

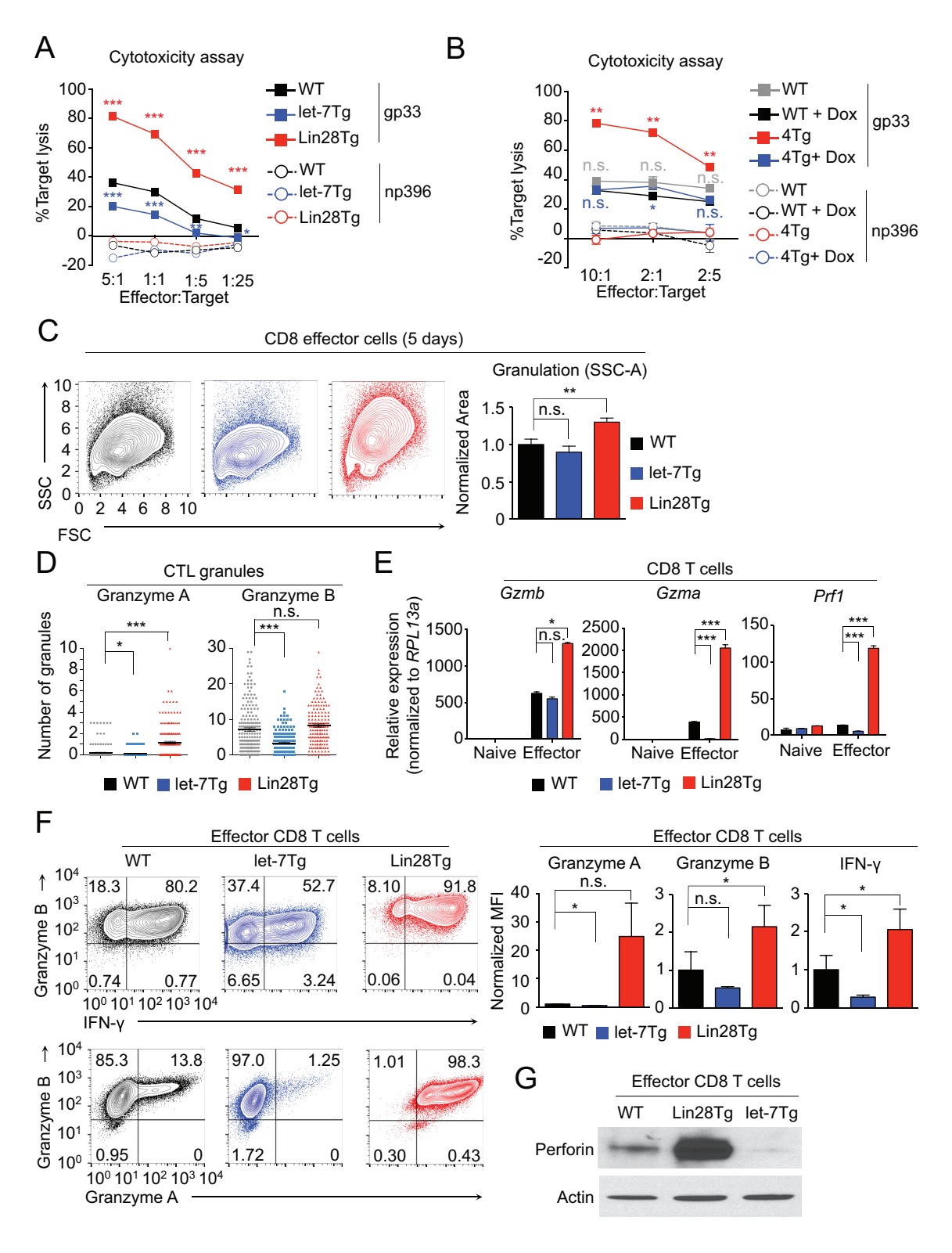

**Figure 4.** let-7 miRNAs negatively regulate differentiation and acquisition of effector function in CTLs. (A, B) Cytotoxicity assay of differentiated CTLs from P14+wild type, P14+let-7Tg, and P14+Lin28Tg lymph nodes (A) or from P14+wild-type (WT), or P14+let-7Tg+Lin28Tg+*Rag2*−/− (4Tg) lymphocytes co-cultured with either LCMV gp33 or LCMV np396 peptide-pulsed splenocytes for 4–5 hr, either in the presence or absence of doxycycline (B). (C) Analysis of the internal complexity (FSC, forward scatter; SSC, side scatter) of effector (5 days after anti-CD3 mAb stimulation) CD8 T cells generated

*Figure 4 continued on next page*

*Figure 4 continued*

from wildtype, let-7Tg, and Lin28Tg lymphocytes (left) and quantification of SSC of CD8 T cells normalized to wild-type. (D) Quantification of Granzyme A and Granzyme B- positive granules in wild-type, let-7Tg, and Lin28Tg CTLs via MilliPore Amnis ImageStream. (E) Quantitative RT-PCR analysis of effector molecule mRNA expression: *Gzma* (Granzyme A), *Gzmb* (Granzyme B), *Prf1* (Perforin) in naive and effector CD8 T cells from wild-type, let-7Tg, and Lin28Tg lymph nodes, presented relative to expression of the ribosomal protein *Rpl13a*. (F) Staining (top, middle) and MFI (bottom) of Granzyme B, Interferon-γ, and Granzyme A in wild-type, let-7Tg, and Lin28Tg effector CD8 T cells normalized to wild type. (G) Western blot analysis of lysates of wild-type, let-7Tg, and Lin28Tg effector CD8 T cells, probed with monoclonal antibodies against Perforin and actin. n.s., not significant (p>0.05), *p<0.05, **p<0.01, and ***p<0.001, compared with wild-type using two-tailed Student's *t*-test. Data are from one experiment representative of three experiments (a, b, e; mean and s.e.m. of technical triplicates, f; mean and s.e.m of three experiments).

The following source data and figure supplement are available for figure 4:

**Source data 1.** Quantification of cytotoxic assays and expression of effector molecules assessed by qPCR and flow cytometry in CTLs.
**Figure supplement 1.** Effector molecule mRNA expression in 4Tg mice.

luciferase activity was restored, confirming that let-7 microRNAs directly target *Eomes* mRNA (*Figure 5E*).

## let-7 miRNAs suppress CD8 T cell function through targeting *Eomes* mRNA

To ascertain whether let-7 controlled CD8 T cell differentiation and function through Eomes, we used a loss-of-function approach to knockout Eomes expression in Lin28Tg CTLs, expecting the reduction of enhanced cytotoxicity in let-7-deficient cells. T-cell specific deletion of either one or both alleles of *Eomes* in CTLs generated from P14$^+$CD4Cre$^+$Eomesfl$^{fl/wt}$ Lin28Tg, and P14$^+$CD4Cre$^+$-Eomesfl$^{fl/fl}$Lin28Tg T cells resulted in gradually decreased levels of Eomes (*Figure 6A*). Any residual Eomes expression in CTLs derived from *CD4Cre$^+$Eomesfl$^{fl/fl}$*Lin28Tg mice, was attributed to 'escapees' of the conditional knockout. The expression of the effector molecules *Gzma*, and *Prf1* was reduced in a manner consistent with the expression of Eomes, where loss of Eomes ameliorated the enhanced expression of these effector molecules in Lin28Tg effector cells (*Figure 6B,C*). Interestingly, the expression of Granzyme B was not affected by loss of Eomes expression, suggesting Granzyme B may be more complexly regulated through other let-7-dependent, but Eomes-independent mechanisms (*Figure 6—figure supplement 1*). Ultimately, the loss of Eomes expression resulted in a significant reduction of cytotoxic function, even in the absence of let-7 microRNAs (*Figure 6D*) while restored Eomes expression completely rescued both the cytotoxic function and the expression of effector molecules in Eomes-deficient Lin28Tg CTLs (*Figure 6—figure supplement 2A,B*). Furthermore, forced expression of the mutated form of Eomes (lacking the let-7-binding motif) in let-7Tg CTLs also enhanced the effector phenotype (*Figure 6—figure supplement 2C,D*). These results demonstrate that let-7-mediated suppression of Eomes is in part responsible for the compromised differentiation and cytotoxic function of let-7Tg CTLs in vitro.

Thus, our data suggest a model in which the let-7 miRNAs act as a molecular control hub that drives CD8 T cell differentiation and function, in a manner dependent on the magnitude of TCR stimulation (*Figure 6—figure supplement 3*). Furthermore, we propose that modulation of let-7 miRNA expression in CD8 T cells provides an exciting, novel therapeutic application to control CTL responses.

## Discussion

Our study has identified a critical role for the let-7 miRNAs in regulating the transition between naive and effector stages of CD8 T cells. Specifically, let-7 expression is high in naive T cells and, when absent, results in increased proliferation and expression of activation markers, which may suggest a loss of the quiescent phenotype in CD8 lymphocytes. Moreover, the entire family of let-7 miRNAs is downregulated upon antigen stimulation through the TCR, which we demonstrated is necessary for the successful progression of CD8 T cell differentiation into CTLs. Therefore, our data suggest that let-7 expression keeps CD8 T cells in a naive state and prevents CTL differentiation, while the

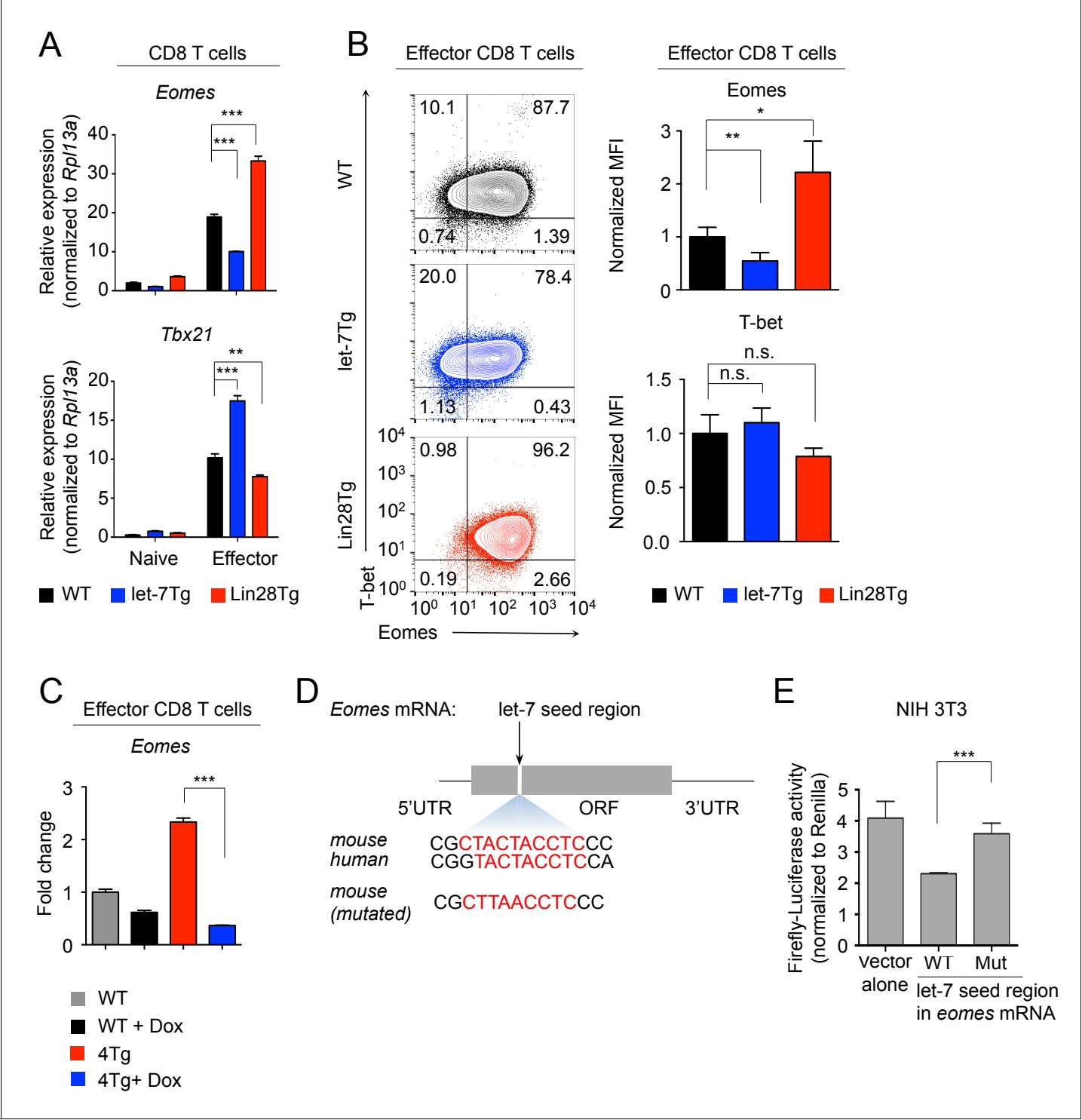

**Figure 5.** let-7 miRNAs directly target the mRNA of *Eomes* in activated CD8 T cells. (**A**) Quantitative RT-PCR analysis of *Eomes* (Eomesodermin) and *Tbx21* (T-bet) mRNA in naive and effector CD8 T cells (5 days after anti-CD3 mAb stimulation) generated from wild-type, let-7Tg, and Lin28Tg lymphocytes, presented relative to the ribosomal protein *Rpl13a*. (**B**) Staining of Eomes and T-bet (left) and MFI of Eomes and T-bet (right) in wild-type, let-7Tg and Lin28Tg effector CD8 T cells, normalized to wild-type. (**C**) Quantitative RT-PCR analysis of *Eomes* (Eomesodermin) mRNA in P14[+]wild-type, or P14[+]let-7Tg[+]Lin28Tg[+]*Rag2*[−/−] (4Tg) lymph nodes cultured either in the presence or absence of Dox, presented as the fold change in expression, normalized to wild-type. (**D**) *Eomes* mRNA, including the 5' and 3' untranslated regions (UTR) and the open-reading frame (ORF) within which one let-7-binding site was identified (top), sequence conservation between the mouse and human let-7-binding sites of Eomes is shown (middle). Mutated sequence of the let-7-binding site in the cDNA of mouse Eomes (bottom) used in the luciferase reporter assay. (**E**) Luciferase reporter assay of let-7

*Figure 5 continued on next page*

*Figure 5 continued*

targeting of the Eomes ORF in NIH 3T3 fibroblasts transfected with a luciferase reporter containing either the intact or mutated sequence of the let-7 seed region from the mouse Eomes ORF; the activity of firefly luciferase was normalized to the Renilla luciferase control. n.s., not significant (p>0.05), *p<0.05, **p<0.01, and ***p<0.001, compared with wild-type using two-tailed Student's *t*-test. Data are from one experiment representative of three independent experiments (a, b, c, e; mean and s.e.m. of technical triplicates (**A,C,E**) or three experiments (**B**)).

The following source data and figure supplements are available for figure 5:

**Source data 1.** Quantification of luciferase activity, and the expression of transcription factors in CTLs assessed by flow cytometry and qPCR.
**Figure supplement 1.** Effect of let-7 miRNAs on expression of transcription factors in effector CD8 T cells.
**Figure supplement 2.** Let-7 expression and luciferase detection in NIH 3T3 fibroblasts.

magnitude of TCR-mediated downregulation of let-7 expression guides the proliferation, differentiation and the acquisition of effector function of CD8 T cells.

The importance of miRNAs in the regulation of the immune system has been demonstrated through the deletion of the miRNA processing enzyme, Dicer. In fact, it has been shown that Dicer deficiency promotes the differentiation of CD8 T cells into CTLs (*Trifari et al., 2013*), suggesting the involvement of specific miRNAs.

Let-7 miRNAs, the largest and most abundantly expressed family of miRNAs in CD8 T cells, have been shown to be important in early development, metabolism and cancer (*Zhu et al., 2011*; *Abbott et al., 2005*; *Büssing et al., 2008*; *Roush and Slack, 2008*; *Yu et al., 2007*). Recent studies have also implicated the importance of the let-7 miRNAs in the development, maintenance and function of the immune system, including T cells (*Pobezinsky et al., 2015*; *Yuan et al., 2012*; *Okoye et al., 2014*). Using both gain-of-function experiments employing the let-7Tg mouse capable of sustaining let-7 expression following TCR activation, and loss-of-function experiments using the Lin28Tg mouse, where the Lin28Tg inhibits let-7 expression, our study has corroborated these earlier reports, and has identified a novel role for let-7 in CD8 T cell differentiation and function.

Prior to antigen encounter, naive CD8 T cells are maintained in a quiescent state, in which T cells have no effector function, are metabolically inactive and undergo minimal proliferation. It has been shown that the homeostasis of naive T cells depends on the balance of two signals, the recognition of low-affinity self-ligands by the TCR, and cytokine stimulation (*Goldrath et al., 2002*; *Kamimura and Bevan, 2007*; *Kieper et al., 2004*, *Kieper et al., 2002*; *Kimura et al., 2013*). Although it has become clear that the weak recognition of self-ligands is not enough for naive T cells to lose the quiescent state, the molecular mechanism that prevents the spontaneous activation of T cells under these conditions is not fully understood. Here, we have demonstrated that the high expression of let-7 miRNAs is necessary to maintain CD8 T cells in a naive state. In fact, we observed that let-7 ablation in CD8 T cells leads to the loss of the quiescent phenotype, as indicated by more active proliferation, an overall increase of cell size, and the upregulation of activation markers such as CD122, and CD44 (*Cuylen et al., 2016*; *Intlekofer et al., 2005*). These experiments were conducted using CD8 T cells from P14$^+$Lin28Tg*Rag2*$^{-/-}$ mice in order to prevent bystander effects from IL-4-producing PLZF$^+$ cells present in Lin28Tg mice (*Pobezinsky et al., 2015*; *Yuan et al., 2012*; *Weinreich et al., 2010*). Although further investigation is needed to distinguish between the let-7-dependent and let-7-independent effects of the Lin28 transgene in naive CD8 T cells, we can speculate that high expression of let-7 miRNAs prevents the spontaneous activation of naive T cells. Furthermore, it will be interesting to explore the precise let-7-mediated regulation of transcription programs, such as Foxp1 (*Feng et al., 2011*), that may contribute to the control of CD8 T cell quiescence.

Furthermore, we have found that the let-7-mediated 'molecular brake' is released upon antigen stimulation, due to the profound downregulation of all members of the let-7 miRNA family in activated CD8 T cells in response to TCR stimulation, in a manner proportional to its strength. Using in vivo models to assess both antiviral and antitumor immunity, we have demonstrated the importance of let-7 miRNAs in controlling CD8 T cell-mediated immune responses. Let-7Tg CD8 T cells failed to proliferate in response to acute LCMV infection, and the few let-7Tg cells that did respond exhibited

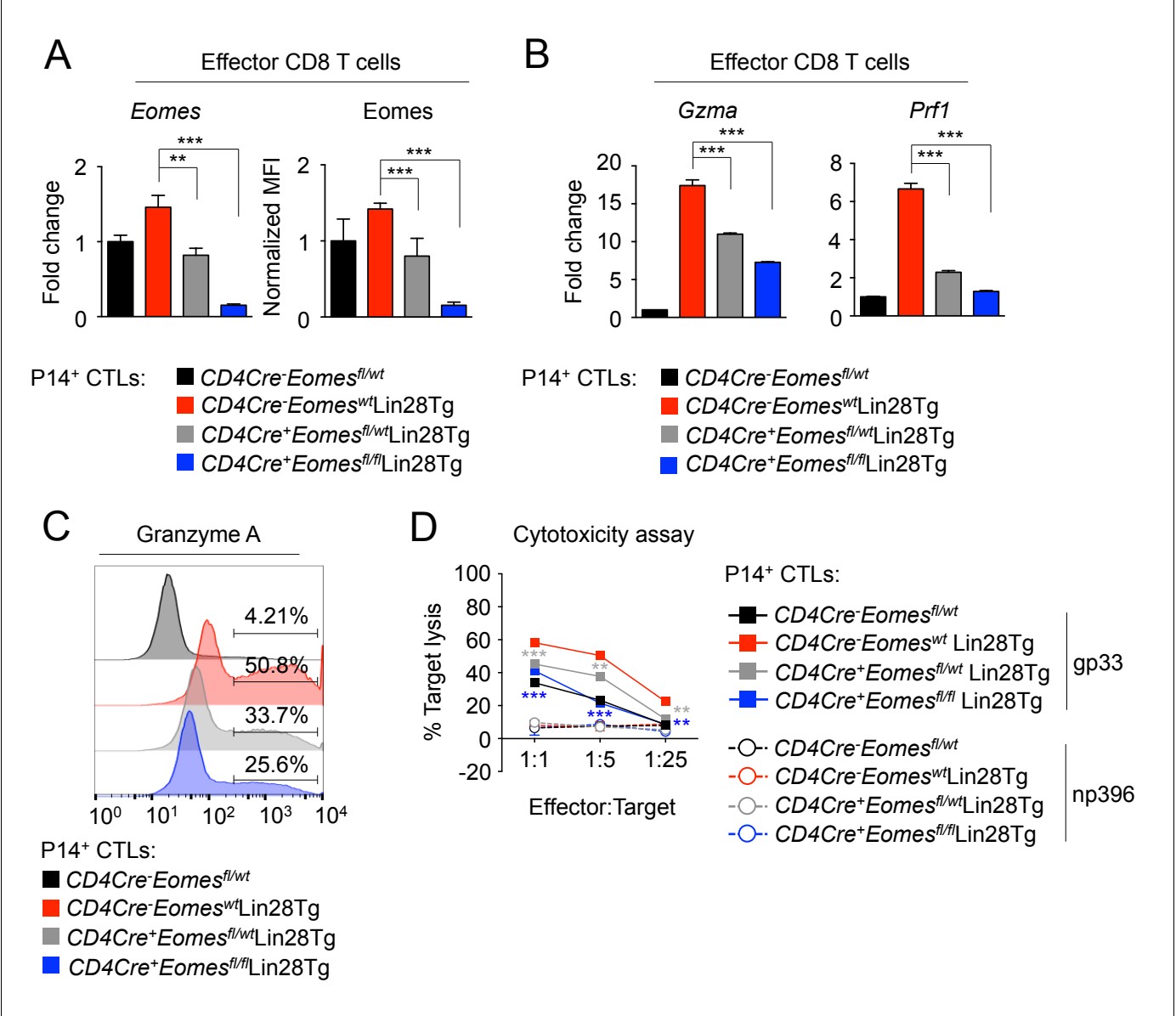

**Figure 6.** let-7 miRNAs control the differentiation of CTLs through Eomes-dependent and independent mechanisms. (**A**) Quantitative RT-PCR analysis of *Eomes* mRNA, presented as the fold change in expression, normalized to wild-type (left) and MFI of Eomes protein expression, normalized to wild-type (right), from CTLs generated from P14⁺*CD4Cre⁻Eomes*fl/wt, P14⁺*CD4Cre⁻Eomes*wtLin28Tg, P14⁺*CD4Cre⁺Eomes*fl/wtLin28Tg, and P14⁺*CD4Cre⁺Eomes*fl/flLin28Tg CD8 T cells. (**B**) Quantitative RT-PCR analysis of *GzmA* (Granzyme A) and *Prf1* (Perforin) mRNA in effector CTLs generated from the indicated mice, presented as the fold change in expression, normalized to wild-type. (**C**) Staining of Granzyme A in CTLs generated from the indicated mice. (**D**) Cytotoxicity assay demonstrating specific target lysis of differentiated P14⁺ CTLs from the indicated mice, co-cultured with either LCMV gp33 or LCMV np396 peptide-pulsed splenocytes for 4–5 hr. **p<0.01, and ***p<0.001, compared with Lin28Tg using two-tailed Student's *t*-test. Data are from one experiment representative of two independent experiments (a; mean and s.e.m. of two experiments; b, d; mean and s.e.m. or technical triplicates).

The following source data and figure supplements are available for figure 6:

**Source data 1.** Quantification of cytotoxic assays and expression of effector molecules assessed by qPCR and flow cytometry in CTLs.

**Figure supplement 1.** Granzyme B expression is regulated independent of Eomes.

**Figure supplement 2.** Re-expression of Eomes enhances cytotoxic function of CTLs.

*Figure 6 continued on next page*

*Figure 6 continued*

**Figure supplement 3.** Model of let-7- mediated regulation of CD8 T cell differentiation.

a very weak effector phenotype, suggesting that downregulation of let-7 is critical to both the proliferative burst of antigen-specific CD8 T cells upon encounter with viral antigen, and the differentiation of these cells in vivo. These results were further bolstered by the failure of let-7Tg mice to reject an allogeneic tumor, the P815 mastocytoma. Altogether, our in vivo studies demonstrate the significance of let-7 downregulation in effector CD8 T cells, and suggest a novel level of control of immune responses that can be therapeutically targeted for treatment of infectious diseases, cancer and autoimmunity.

Antigen stimulation of T cells results in the increased biosynthesis that is needed to support the clonal expansion of antigen specific lymphocytes, and ultimately the acquisition of effector function. The hallmark of this process is a metabolic switch from oxidative phosphorylation to aerobic glycolysis (*Frauwirth et al., 2002*; *Frauwirth and Thompson, 2004*), as well as a concomitant increase in cell size, both of which have been reported to be controlled by Myc (*Chou et al., 2014*; *Wang et al., 2011*; *Grumont et al., 2004*), the expression of which is rapidly induced upon antigen stimulation of CD8 T cells (*Williams and Bevan, 2007*; *Wang et al., 2011*; *Grumont et al., 2004*). Presumably due to the pro-apoptotic activity of Myc (*Askew et al., 1993*; *Evan et al., 1992*; *Fanidi et al., 1992*), its expression is transient, eventually receding during the later stages of CD8 T cell differentiation (*Best et al., 2013*; *Nie et al., 2012*). In our study, we have shown that let-7 suppresses the expression of Myc on the mRNA level, and consequently modulates the function of Myc in CD8 T cells, based on our assessment of established Myc targets, supporting the previous observation of Myc as a non-canonical target of let-7 miRNAs in cancer cells (*Kim et al., 2009*; *Sampson et al., 2007*). Our results suggest that let-7 likely regulates the metabolic switch in activated CD8 T cells through Myc.

We also found that the let-7 miRNAs may directly inhibit the proliferation of activated CD8 T cells by suppressing the expression of the cell cycle regulators Cdc25a, Ccnd2 and Cdk6, all of which are known let-7 targets (*Johnson et al., 2007*). Since these factors have also been described as transcriptional targets of Myc (*Bouchard et al., 1999*; *Galaktionov et al., 1996*; *Mateyak et al., 1999*), we cannot rule out the possibility of more complex regulation where let-7 is not solely responsible for controlling their expression.

Moreover, we demonstrated that let-7-mediated suppression of CD8 T-cell immune responses is also due to modulation of effector function. We noticed that the internal complexity of CTLs generated in vitro was diminished in the presence of forced let-7 expression, and subsequently determined that this was due to a reduction in the number of cytotoxic granules, as well as in the expression of effector molecules. We thus concluded that let-7 functions as a molecular rheostat that quantifies TCR signaling to direct the CTL response upon antigen stimulation, in a similar fashion to other miRNAs, including miR-181 and the miR-17–92 cluster (*Henao-Mejia et al., 2013*; *Li et al., 2007*; *Wu et al., 2012*). These conclusions are supported by a recent study demonstrating that in neonatal mice the residual expression of Lin28B in T lymphocytes skews CD8 T cell differentiation toward an effector phenotype (*Wang et al., 2016*).

We further wanted to determine the molecular mechanisms through which let-7 acts to inhibit CTL differentiation. We found that *Eomes* is directly regulated by let-7 miRNAs, which can in part explain the block in the differentiation of CD8 T cells with forced let-7 expression. Yet, in contrast to previous studies on the differential roles of Eomes and T-bet in governing CD8 T cell function, we have shown that heightened expression of Eomes in effector T cells may be more important for effector function than previously thought (*Pearce et al., 2003*; *Intlekofer et al., 2005*; *Chang et al., 2014*). In fact, let-7/Eomes double-deficient CTLs had reduced antigen-specific cytotoxicity in vitro. However, our data also suggest that Eomes is only a part of this 'cytotoxic program', as deletion of Eomes only reduced cytotoxicity to the levels exhibited by wild-type CTLs, and re-expression of Eomes in let-7Tg CTLs only marginally increased cytotoxicity. These results indicate that other let-7-dependent, but Eomes-independent mechanisms are at play. More experiments are required to elucidate these Eomes-independent mechanisms, and should be enlightening given the suggested

redundancy of Eomes activity. Additionally, we have identified Granzyme A as a probable target of Eomes, while also confirming that Perforin, and IFN-γ expression is controlled by Eomes (*Pearce et al., 2003*). These results are consistent with the enhanced cytotoxicity exhibited by Lin28Tg CD8 T cells with increased Eomes expression. We also observed previously reported reciprocal expression between T-bet and Eomes (*Intlekofer et al., 2005*; *Chang et al., 2014*; *Nayar et al., 2012*).

These results clearly demonstrate that downregulation of let-7 upon TCR stimulation is a critical process in the determination of the magnitude of the CD8 T cell response in vivo, as let-7 miRNAs inhibit proliferation and differentiation by targeting cell cycle regulators, affecting metabolic reprogramming through the suppression of Myc, and repression of effector function through Eomes-dependent and -independent mechanisms. Thus, naive CD8 T cells require let-7 miRNAs to remain quiescent, and only upon antigen stimulation through the TCR can this molecular 'brake' be released. Based on these results, we propose that let-7 miRNAs act as a molecular control hub that translates TCR signaling to control CD8 T cell differentiation and function.

## Materials and methods

### Animals

C57BL/6J (CD45.2+wild type, stock no. 000664), B6.SJL- *Ptprc$^a$Pepc$^b$*/BoyJ (CD45.1+wild type, stock no. 002014), B6(Cg)- *Rag2$^{Tm1.1Cgn}$*/J (*Rag2$^{-/-}$*, stock no. 008449), B6 Tg(CD4-cre)1Cwi/BfluJ (*CD4-Cre*, stock no. 017336), and B6.129S1 (Cg)- *Eomes$^{tm1.1Bflu}$*/J (*Eomes$^{fl/fl}$*, stock no. 017293) were obtained from the Jackson Laboratory. B6.Cg- *Col1a1$^{tm3(tetO-Mirlet7g/Mir21)Gqda}$*/J (let-7g, stock no. 023912) and B6.Cg- *Gt(ROSA)26 Sor$^{tm1(rtTA*M2)Jae}$*/J (M2rtTA, stock no. 006965) were also obtained from the Jackson Laboratory and subsequently crossed to generate let-7Tg mice. Mice with the Lin28B transgene (Lin28Tg) driven under the control of the human CD2 promoter (*Pobezinsky et al., 2015*), and B6 Tg(TcrLCMV)327Sdz/JDvs/J (P14) mice were a generous gift from Alfred Singer (NCI, NIH). P14+ mice, and wild-type C57Bl/6J mice on a *Rag2$^{-/-}$* background were crossed to generate wild-type P14+ mice. Let-7Tg mice, and P14+ mice were crossed on a *Rag2$^{-/-}$* background to generate P14+ doxycycline-inducible let-7 transgenic mice. Lin28Tg mice were crossed with P14+ mice on a *Rag2$^{-/-}$* background to generate P14+Lin28Tg mice. let-7Tg, Lin28Tg, P14+ mice were crossed on a *Rag2$^{-/-}$* background to generate 4Tg mice. *CD4Cre* and *Eomes$^{fl/fl}$* mice were crossed to generate mice with a T-cell-specific conditional knockout of Eomes. *CD4Cre*, *Eomes$^{fl/fl}$*, Lin28Tg, and P14+ mice were crossed to generate Lin28Tg mice with a T-cell-specific deletion of Eomes (P14+*CD4Cre*+*Eomes$^{fl/fl}$*Lin28Tg). Littermates or age and sex-matched mice were used as controls. All breedings were maintained at the University of Massachusetts, Amherst. This study was performed in accordance with the recommendations in the Guide for the Care and Use of Laboratory Animals of the National Institutes of Health. All the animals were handled according to approved institutional animal care and use committee (IACUC) protocols (#2014–0045, 2014–0065, 2015–0035) of the University of Massachusetts.

### Doxycycline-mediated induction of let-7 transgene expression

Experimental mice including control animals (unless specifically stated otherwise) were fed with 2 mg/mL doxycycline in drinking water supplemented with 10 mg/mL sucrose for 4 days prior to the initiation of experimental procedures to ensure maximal induction of let-7g expression. Doxycycline drinking water was replaced every other day. In vitro, lymphocytes were cultured with 2 µg/mL doxycycline in CTL culture media (see cell sorting and in vitro culture below).

### In vivo BrdU labeling

Mice were injected i.p. with 1 mg BrdU in PBS, and subsequently fed with 0.8 mg/mL BrdU in drinking water supplemented with 2% sucrose for 4 days. BrdU water was kept in the dark to eliminate light-sensitivity effects of BrdU and was replaced daily. Incorporation of BrdU in CD8 T cells from the spleen and lymph nodes were analyzed by flow cytometry (see flow cytometry analysis below).

## Flow cytometry analysis

Flow cytometry data were acquired on a BD Fortessa or a MilliPore Amnis ImageStream. The following monoclonal antibodies were used: CD8$\alpha$ (53–6.7, eBioscience, RRID:AB_962672; 5H10, Invitrogen, RRID:AB_10372364), CD8$\beta$ (YTS156.7.7, BioLegend, RRID:AB_961293), CD4 (RM4-5, BioLegend, RRID:AB_312718), BrdU (3D4, BioLegend, RRID:AB_2564481), CD44 (IM7, BD Pharmingen, RRID:AB_2076224), CD25 (PC61, BioLegend, RRID:AB_312857; PC61.5 eBioscience, RRID:AB_465606), CD122 (TM-$\beta$1, BioLegend, RRID:AB_940607), Granzyme A (3G8.5 BioLegend, RRID:AB_2565309; eBioscience, RRID:AB_2573227), Granzyme B (GB11, BioLegend, RRID:AB_2562195), IFN-$\gamma$ (XMG1.2, eBioscience, RRID:AB_469503), Eomes (Dan11mag, eBioscience, RRID:AB_1603275), Ki67 (SolA15, eBioscience, RRID:AB_11149672), T-bet (O4-46, BD Pharmingen, RRID:AB_10564093), PE-Streptavadin (BioLegend, RRID:AB_2571914).

For restimulation in vitro, $2 \times 10^6$ cells were stimulated with phorbol myristate acetate (PMA) and Ionomycin for 4 hr at 37°C and Monensin A for 2 hr at 37°C. Cells were first stained for surface proteins then fixed, permeablized, and stained for intracellular proteins according to the manufacturer's instructions (BD Pharmingen, eBioscience).

For LCMV studies, $2 \times 10^6$ cells were stimulated for 4 hr at 37°C with 2 µg/mL of GP$_{33-41}$ or NP$_{396-404}$ peptides, and 1 µl/mL GolgiPlug (BD Pharmingen). Cells were first stained for surface proteins then fixed, permeablized, and stained for intracellular proteins according to the manufacturer's instructions (BD Pharmingen). The following monoclonal antibodies were used: CD8$\beta$ (YTS156.7.7; BioLegend; RRID:AB_961293), CD45.1 (A20; BD Pharmingen; RRID:AB_1727488), CD45.2 (104; BD Pharmingen; RRID:AB_1727491), KLRG1 (2F1; BD Pharmingen; RRID:AB_393931), CD44 (IM7; BD Pharmingen; RRID:AB_2076224), TNF-$\alpha$ (MP6-XT22; BD Pharmingen; RRID:AB_395380), IFN-$\gamma$ (XMG1.2; BD Pharmingen; RRID:AB_398551). Flow cytometry data were acquired on a BD LSR II. All flow cytometry data were analyzed with FlowJo software (TreeStar; RRID:SCR_008520). MilliPore Amnis ImageStream data were analyzed with IDEAS software (EMD Millipore).

## Western blot analysis

Cells were collected and lysed in M2 lysis buffer (20 mM Tris, pH7.0, 0.5% NP40, 250 mM NaCl, 3 mM EDTA, 3 mM EGTA, 2 mM DTT, 05 mM PMSF, 20 mM $\beta$-glycerol phosphate, 1 mM sodium vanadate, 1 µg/mL Leupeptin), then resolved by SDS-PAGE. Blots were probed with anti-Perforin (CB5.4, Abcam; RRID:AB_302236) and anti-actin (AC40, Sigma; RRID:AB_476730), and visualized using enhanced chemiluminescence (ThermoScientific) with horse-radish peroxidase conjugated anti-rat IgG (712-035-150, Jackson ImmunoResearch; RRID:AB_2340638) or anti-mouse IgG (401215, Calbiochem; RRID_AB:2686924).

## Cell sorting and in vitro culture

Lymph nodes were harvested and gently tweezed to remove lymphocytes. CD8 lymph node T cells were enriched for via antibody-mediated depletion of B cells using anti-mouse IgG magnetic beads (BioMag, Qiagen). CD4 T cells were removed via anti-rat IgG magnetic beads (BioMag, Qiagen) following incubation with anti-mouse CD4 antibodies conjugated with rat IgG (GK1.5). Lymphocytes were electronically sorted for the further purification of naive CD8 T cells (CD44$^{lo}$ CD25$^{lo}$CD8$^+$CD4$^-$) (*Figure 3—figure supplement 1A*).

Cells were stimulated either with irradiated splenocytes loaded with anti-CD3 mAbs (10 µg/mL), or plate-bound anti-TCR$\beta$ mAbs (10 µg/mL) and anti-CD28 mAbs (5 µg/mL), then differentiated for 5 days in RPMI supplemented with 10% fetal bovine serum, 1% HEPES, 1% sodium pyruvate, 1% penicillin/streptomycin, 1% L-glutamine, 1% non-essential amino acids, 0.3% $\beta$-mercaptoethanol, 100 U/mL IL-2, 100 mg/mL gentamicin, and 2 µg/mL doxycycline when necessary.

## Eomes site-directed mutagenesis and retroviral transduction

Site-directed mutagenesis of the let-7 binding site in the Eomes ORF was performed using the Agilent QuikChange II Site-Directed Mutagenesis kit according to the manufacturer's protocol.

Retrovirus expressing Eomes or Eomes$^{MUTANT}$ cDNA with a GFP reporter were produced from the transfection of PlatE cells using Lipofectamine plus (Invitrogen). For retroviral transduction, naive lymphocytes were stimulated with irradiated splenocytes in the presence of anti-CD3 mAbs (10 µg/

mL) for 24 hr, then spin-fected (660 g, 90 min, 30°C) with virus and polybrene (4 µg/mL). Retrovirally transduced cells were obtained by sorting on the GFP⁺ population.

## Prediction of miRNA target

Eomes was independently identified in an unbiased search of all ORFs in the mouse and human genomes, for matches to an extended 9 bp let-7 seed, 'TACTACCTC'. This search utilized a hashing algorithm as described in (*Markstein et al., 2002*) and identified 119 genes in the mouse genome, and 159 genes in the human genome that have one or more matches to the 9 bp let-7 seed in their ORF sequences. Interestingly, humans have three splice variants of Eomes, one of which lacks the exon containing the match to let-7, thus opening the possibility that Eomes may escape let-7 repression in some cells by alternative splicing of the target sequence. This may require further investigation.

## Luciferase assay

NIH 3T3 cells (ATCC) were transfected with the pmirGLO vector (Promega) containing either the intact let-7 binding motif from Eomes, or a mutated version of this binding motif, or either intact antisense or mutated antisense seed regions of let-7b, let-7g, or let-7i using Lipofectamine and Plus reagent (Invitrogen). Luciferase activity was measured 48 hr later on a POLARstar Omega 96-well plate reader (BMG Labtech), using the Dual-Luciferase Reporter Assay System (Promega).

## CellTrace violet proliferation assay

Electronically sorted naive CD8 T cells were labeled with CellTrace Violet (Invitrogen) for 15 min at 37°C. Cells were stimulated using plate-bound anti-TCRβ mAbs (10 µg/mL) and anti-CD28 mAbs (5 µg/mL), cultured for 72 hr, and analyzed by flow cytometry.

## CellTrace violet cytotoxicity assay

P14⁺ CD8 T cells were stimulated with anti-TCRβ mAbs (10 µg/mL) and anti-CD28 mAbs (5 µg/mL) plate-bound antibodies, differentiated into CTLs for 5 days in the presence of IL-2, gentamicin, and 2 µg/mL doxycycline when appropriate. On day 5, live splenocytes were warmed for 10 min at 37°C, then labeled with CellTrace Violet (Invitrogen) at two different concentrations (CTV^high or CTV^low) for 15 min at 37°C. CTV^low splenocytes were then loaded with either LCMV gp33-41 peptide (1 µM, GenScript) or LCMV np396-404 peptide (1 µM, GenScript) for 1 hr at 37°C, and are referred to below as 'experimental splenocytes'. CTV^high splenocytes remained peptide-free, and were used as a reference control, referred to below as 'control splenocytes'. Equal amounts of both experimental and control splenocytes were co-cultured with CTLs at different ratios for 4 to 5 hr. Cytotoxicity was assessed by flow cytometry using propidium iodide.

Measured as the percent target lysis of live experimental splenocytes loaded with either target (gp33-41) or control (np396-404) peptide from the lymphocytic choriomeningitis virus. The following formula was used to calculate the percent target lysis:

$$\left(1 - \left(\frac{A}{B}X\frac{C}{D}\right)\right)X\,100$$

A- frequency of live experimental splenocytes co-cultured with CTLs; B- frequency of live control splenocytes co-cultured with CTLs; C- frequency of live control splenocytes incubated in the absence of CTLs; D- frequency of live experimental splenocytes incubated in the absence of CTLs.

## Lymphocytic choriomeningitis virus infection and T cell adoptive transfer

10 × 10³ CD45.2⁺ P14⁺ donor cells from the indicated mice were transferred i.v. into CD45.1⁺ congenic hosts. Mice were inoculated with 5 × 10⁴ p.f.u. of LCMV Armstrong i.p.. Spleens were harvested and processed 7 days post- infection.

## Isolation of RNA and quantitative PCR

RNA was isolated according to the manufacturer's instructions (QIAGEN miRNeasy), and genomic DNA removed using the DNA-free DNA removal kit (Ambion). mRNA-encoding cDNA was

synthesized using the SuperScript III Reverse Transcriptase kit (Invitrogen), while miRNA- encoding cDNA was synthesized using the Taqman MicroRNA Reverse Transcription kit (Applied Biosystems). SYBR Green quantitative PCR was performed using the Bioline SensiFAST SYBR Lo-Rox kit and Taqman quantitative PCR was performed using the Bioline SensiFAST Lo-Rox kit. Both SYBR Green and Taqman amplification primers (Integrated DNA Technologies, or Applied Biosystems) are listed in *Supplementary file 2*.

## Tumor transplantation

For P815 (ATCC; RRID:CVCL_2124) survival studies, $30 \times 10^6$ tumor cells were injected i.p. into host mice. For P815 tumor burden studies, $20 \times 10^6$ tumor cells were injected i.p. into host mice. Seven days post-injection, tumor burden was assessed by washing the peritoneum with cold PBS and counting the collected cells. For studies involving let-7Tg mice, all mice were fed with doxycycline for the duration of the study. For studies involving adoptive CD8 T cell transfer, $10 \times 10^6$ naïve CD8 T cells were injected i.v. into $P14^+ Rag2^{-/-}$ mice, concurrently with i.p. tumor injection. The P815 cell line was authenticated by analyzing H-2K$^d$ expression via flow cytometry.

## Statistical analysis

*p*-Values were determined using a two-tailed Student's t-test, or one-tailed Student's t-test where indicated.

## Acknowledgements

We thank A Singer, X Tai, T Kadakia, F Van Laethem, M Y Kimura, and B A Osborne for critical reading of the manuscript; A Singer for providing reagents, and Lin28Tg and P14 mice; V Lazarevic for providing the plasmid encoding Eomes-GFP and control empty retrovirus; P Markstein and E D'Souza for assistance with the computational analysis of predicted miRNA targets; and G Abbruzzesse for technical help.

## Additional information

### Funding

| Funder | Grant reference number | Author |
| --- | --- | --- |
| National Multiple Sclerosis Society | PP-1503-03417 | Leonid A Pobezinsky |
| University of Massachusetts Amherst | Start up funds | Leonid A Pobezinsky |

The funders had no role in study design, data collection and interpretation, or the decision to submit the work for publication.

### Author contributions

ACW, Conceptualization, Data curation, Formal analysis, Validation, Investigation, Visualization, Methodology, Writing—original draft, Writing—review and editing; KAD, Resources, Data curation, Methodology; CCA, Data curation, Formal analysis, Validation, Writing—review and editing; EF, Data curation, Validation, Writing—review and editing; ASB, Resources, Formal analysis, Methodology; MM, Resources, Software, Validation, Writing—review and editing; DA, Resources, Formal analysis, Supervision, Methodology, Writing—review and editing; RMW, Conceptualization, Resources, Supervision, Funding acquisition, Investigation, Methodology, Writing—review and editing; ELP, Conceptualization, Data curation, Formal analysis, Supervision, Funding acquisition, Investigation, Visualization, Methodology, Project administration, Writing—review and editing; LAP, Conceptualization, Resources, Data curation, Formal analysis, Supervision, Funding acquisition, Investigation, Methodology, Writing—original draft, Project administration, Writing—review and editing

Author ORCIDs

Alexandria C Wells, http://orcid.org/0000-0001-9799-8353
Dominique Alfandari, http://orcid.org/0000-0002-0557-1246
Leonid A Pobezinsky, http://orcid.org/0000-0002-6115-3559

Ethics

Animal experimentation: This study was performed in accordance with the recommendations in the Guide for the Care and Use of Laboratory Animals of the National Institutes of Health. All of the animals were handled according to approved institutional animal care and use committee (IACUC) protocols (#2014-0045, 2014-0065, 2015-0035) of the University of Massachusetts.

## Additional files

### Supplementary files

• Supplementary file 1. A list of all genes containing a 9-base pair let-7 binding motif in either the 3' untranslated region (UTR) or open reading frame (ORF) from both the mouse and human genomes, where a (+) or (-) demonstrates the orientation of the gene in the genome. The ending models are an illustration of the location of the let-7 binding motifs within the human and mouse *Eomes* gene, where red represents exons and pink represents introns.

• Supplementary file 2. Primer sequences and Taqman Assay catalog numbers.

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
