## [Decision Letter]

Thank you for submitting your article "Modulation of let-7 miRNAs controls the differentiation of effector CD8 T cells through regulation of Myc and Eomes" for consideration by *eLife*. Your article has been reviewed by two peer reviewers, and the evaluation has been overseen by a Reviewing Editor and Tadatsugu Taniguchi as the Senior Editor. The reviewers have opted to remain anonymous.

The reviewers have discussed the reviews with one another and the Reviewing Editor has drafted this decision to help you prepare a revised submission.

You set out to demonstrate that the let-7 family of microRNAs play a role in differentiation of mouse cytotoxic CD8 T cells, and propose Myc and Eomesdermin (Eomes) as direct down-stream targets. Overall, both reviewers feel that the work was well done, but that some improvements are required to strengthen your conclusions.

Essential revisions:

1) Can the Lin28 effect be overcome with Let-7? There is precedence for Let-7-independent roles of Lin28. This would affect central point of their paper.

2) Are effects of Lin28 cell intrinsic? In Figure 2, separate groups of WT and let-7 Tg mice were injected with P815 tumors and the let-7 Tg mice displayed elevated tumor burden and decreased survival. The authors concluded that this was due to CD8 T cell maintenance of let-7 expression, thereby leading to a defective immune response against the tumor. However, from this experimental design, it is impossible to determine whether this is a CD8 T cell intrinsic effect, as let-7g miRNA is maintained in all lymphocytes in the presence of doxycycline. Therefore, forced expression of let-7g may also be impacting the NK or NKT cell response against P815 tumors. Do the authors have any evidence of an impaired CD8 T cell response in this model?

3) Is the mechanism transcriptional vs post-transcriptional? For example, measure Let-7 precursors in nucleus to see if its transcription or processing?

4) Controls are important and will make results more convincing.a) authors did not perform global profiling, but only used U6 snRNA to normalize for Let-7 levels; can they get same conclusion using another quantification methods. Authors should consider differences in cell size or total RNA content per cell. b) Relative quantification of mature let-7 levels were determined. However, the way it is presented, one cannot discern the levels of individual let-7 family members, and whether they are decreasing significantly. Ideally, absolute quantification would be done instead of expression relative to one housekeeping small RNA. It would be reassuring to see at least one other miRNA that is unchanged and one that is increased to show that the effect on let-7 expression is specific and not generally observed for all miRNAs upon CD8 T cell activation. It would be reassuring to see at least one other miRNA that is unchanged and one that is increased. c) For mice, what were WT controls? Were they age, microbiome, sex- matched littermates or not? For Dox experiments, were they also given Dox to control for side effects of drug?

5) Compensation between Eomes and Tbet? This point is also very important since they claim that Eomes is the critical downstream target. They need to strengthen their arguments for this conclusion. Along these lines, can overexpressing Eomes (using synonymous substitution at the seed sequence targeted by let-7) in Eomes fl/fl Lin28 CD4 Cre+ Tg P14 CD8 T cells rescue T cell function?

6) The authors should be careful about making conclusions about quiescence of CD8 cells based on Ki67 alone and other markers (such as CD44 and CD122). Why would CD44 expression be significantly changed in spleen but not lymph node (Figure 1—figure supplement 3)? A direct assay of cell quiescence could be used. A direct assay of cell quiescence should be included in the analysis.

7) Many metabolic genes are differentially expressed in let-7 and lin28 Tg CD8 T cells. Do the authors have any metabolic analyses on a cellular level (such as OCR and ECR measurements with Seahorse) for WT, let-7 and lin28 Tg CD8 T cells that would help substantiate let-7 as a regulator of the metabolic reprograming that occurs in activated CD8 T cells?

[Editors' note: further revisions were requested prior to acceptance, as described below.]

Thank you for resubmitting your work entitled "Modulation of let-7 miRNAs controls the differentiation of effector CD8 T cells" for further consideration at *eLife*. Your revised article has been favorably evaluated by Tadatsugu Taniguchi (Senior editor) and three reviewers, one of whom is a member of our Board of Reviewing Editors.

The manuscript has been improved but there are some remaining issues that need to be addressed before acceptance, as outlined below:

All reviewers felt that the manuscript was improved by the revision. The data within this manuscript are sufficient to support your model wherein T cell receptor-mediated downregulation of let-7 is important to facilitate CD8 T cell clonal expansion and the acquisition of effector function. Moreover, your discovery that this is at least in part due to derepression of the newly identified let-7 target, Eomes, is an important finding.

There are still some minor issues that need to be dealt with prior to publication, but its anticipated that these can be addressed without additional experiments.

You nicely show that mutated Eomes (with a disrupted let-7 binding motif) can increase the cytotoxic function of let-7Tg CTLs. However, a side-by-side comparison using WT-Eomes may have been even more informative. Is this data available? If so this would strengthen conclusions.

In response to point #1, you wrote: "We found that Let-7 re-expression dramatically suppresses the expression of Eomes and all tested effector molecules in Lin28Tg CTLs." However, there is no analysis of Myc expression in 4Tg mice. Furthermore, Rao published that Let-7 targets Dicer mRNA and thus impairs global miRNA processing, which also promotes the CTL program. Do the authors have any data on Myc or Dicer? Do you agree that Dicer is an important target in this regard? See Trifari et al.,2013.

There is still some imprecise language in the paper. You use the term "loss of quiescence" which is not accurate here. BrDU, like Ki67, indicates proliferation; authors should thus comment only on proliferation rather than "loss of quiescence" because the latter suggests a molecular mechanism. For example, it has been reported that Foxp1 is required for enforcing quiescence on naive CD8 T cells, but you don't present any data about that. Similarly, you should not write "exit from the G0 stage of the cell cycle" since they did not have any data about G0 stage of the cell cycle. Furthermore, you like to use the term "let-7 deficiency." You should not encourage readers to believe that Lin28Tg is equivalent to let-7 deficiency because as discussed already Lin28 most likely affects not only let-7.

In the Abstract, the phase "let-7 microRNAs as a novel molecular transmitter of TCR signaling" might be read as the opposite of the phenotype. As indicated in Figure 6—figure supplement 3, let-7 is drawn as a repressor. Thus, rather than a "transmitter", let-7 stops the signal / acts like a resistor in the circuit. The authors may want to find a term that better reflects the direction of the effect.

---

## [Author Response]

Essential revisions:

1) Can the Lin28 effect be overcome with Let-7? There is precedence for Let-7-independent roles of Lin28. This would affect central point of their paper.

We understand the reviewers’ concern and are very happy to address it. Indeed, it has been demonstrated that Lin28 proteins can bind some mRNAs in addition to Let-7 precursors. To test if the observed effects in Lin28Tg CTLs are due to the let-7 deficiency, we have generated P14+Let-7TgLin28TgRag2KO (4Tg) mice where let-7Tg expression can be induced using doxycycline even in the presence of Lin28Tg. In fact, we have found that the forced expression of the let-7Tg in Lin28Tg CTLs dramatically diminishes their cytotoxic activity as it is shown in Figure 4. In addition to these results, we generated a new Figure 4—figure supplement 1 and a new Figure 5 where we analyzed the expression of effector molecules and the transcription factor Eomes in P14+let7TgLin28TgRagKO CTLs in comparison to P14+Lin28TgRagKO CTLs. We found that Let-7 re-expression dramatically suppresses the expression of Eomes and all tested effector molecules in Lin28Tg CTLs. Altogether our data demonstrate that enhanced differentiation of Lin28Tg CTLs is due to the let-7 deficiency. We also revised the text to emphasize our point.

2) Are effects of Lin28 cell intrinsic?

This is a very important point. We have previously published (Pobezinsky et al., 2015) that lin28-mediated let-7-deficiency in T cells causes abnormal development of IL-4-producing PLZF+ innate T lymphocytes. To eliminate any bystander effects and indirect influence of the Lin28Tg on T cell repertoire formation during thymic selection, we have generated P14+Lin28TgRag2-/- mice, where only CD8^+^ T cells with selected specificity were generated and expressed Lin28 proteins (Lin28Tg is driven by hCD2-cassette that ensures T cell specific expression).

In Figure 2, separate groups of WT and let-7 Tg mice were injected with P815 tumors and the let-7 Tg mice displayed elevated tumor burden and decreased survival. The authors concluded that this was due to CD8 T cell maintenance of let-7 expression, thereby leading to a defective immune response against the tumor. However, from this experimental design, it is impossible to determine whether this is a CD8 T cell intrinsic effect, as let-7g miRNA is maintained in all lymphocytes in the presence of doxycycline. Therefore, forced expression of let-7g may also be impacting the NK or NKT cell response against P815 tumors. Do the authors have any evidence of an impaired CD8 T cell response in this model?

We absolutely agree with the reviewers. To address this critical comment and directly demonstrate that the rejection of the allogeneic P815 tumor in C57BL/6 mice is CD8 dependent, we conducted an additional experiment that is shown in a new Figure 2—figure supplement 1. We demonstrated that the survival of P815 (H-2d) tumor bearing Rag2-/-(H-2b) mice was compromised without adoptive transfer of enriched polyclonal population of CD8 T cells (H-2b). Conversely, Rag2-/- mice that received CD8 T cells were fully protected. Thus, our results and the previously published data (Zhan et al., 2000), clearly demonstrated that in WT(H-2b) mice the rejection of allogeneic P815 tumors is CD8 dependent, justifying the appropriate usage of this model in our work. We also changed the text for clarity and added a new citation.

3) Is the mechanism transcriptional vs post-transcriptional? For example, measure Let-7 precursors in nucleus to see if its transcription or processing?

Excellent point. In fact, we are actively investigating this matter and our very preliminary data suggest that let-7 expression in T cells is regulated on the post-transcriptional level upon TCR-signaling. However, we think that discussion of this mechanism here will dilute the focus of our paper, which is about let-7 mediated regulation of CD8 differentiation and function, but not let-7 biogenesis. No doubt, it is an important question that requires further detailed investigation.

4) Controls are important and will make results more convincing.a) authors did not perform global profiling, but only used U6 snRNA to normalize for Let-7 levels; can they get same conclusion using another quantification methods. Authors should consider differences in cell size or total RNA content per cell.

In response to the reviewer comment, we confirmed let-7 downregulation in activated CD8 T cells using another widely used and suggested housekeeping control RNA for miRNA normalization, snoRNA-135. These results are shown in a new Figure 1—figure supplement 2. Additionally, it has been documented already that effector CD8 T cells express lower levels of let-7 miRNAs in comparison to naïve cells (Wu H. et al., PLoS One 2007).

*b) Relative quantification of mature let-7 levels were determined. However, the way it is presented, one cannot discern the levels of individual let-7 family members, and whether they are decreasing significantly. Ideally, absolute quantification would be done instead of expression relative to one housekeeping small RNA. It would be reassuring to see at least one other miRNA that is unchanged and one that is increased to show that the effect on let-7 expression is specific and not generally observed for all miRNAs upon CD8 T cell activation. It would be reassuring to see at least one other miRNA that is unchanged and one that is increased.*

We agree. In a new Figure 1—figure supplement 1 we display the expression levels and statistics for each individual let-7 miRNA from the experiments shown on the Figure 1 and B. In addition, we used snoRNA135 for normalization to validate our results (new Figure 1—figure supplement 2). We also added expression of a control miRNA, miR-17 that is induced upon TCR-stimulation in CD8 T cells (Katz et al., 2014) (please see a new Figure 1—figure supplement 2).

*c) For mice, what were WT controls? Were they age, microbiome, sex- matched littermates or not? For Dox experiments, were they also given Dox to control for side effects of drug?*

We absolutely agree and apologize for the lack of these details. We clarified this issue in Materials and methods. Briefly, for most of our experiments we used littermates as controls. Age and sex matched C57BL/6 (control) mice were also used when it was appropriate. For “DOX” experiments, we fed ALL mice involved including controls.

5) Compensation between Eomes and Tbet? This point is also very important since they claim that Eomes is the critical downstream target. They need to strengthen their arguments for this conclusion. Along these lines, can overexpressing Eomes (using synonymous substitution at the seed sequence targeted by let-7) in Eomes fl/fl Lin28 CD4 Cre+ Tg P14 CD8 T cells rescue T cell function?

We thank reviewers for this suggestion. In Figure 6 we demonstrated that derepression of Eomes in let-7 deficient CTLs causes enhanced cytotoxic function of CTLs. Specifically, we conditionally knocked out Eomes in let-7 deficient CTLs, which reduced the superior cytotoxic function of these cells. Additionally, to address reviewers comment, we overexpressed Eomes (wild type, un-mutated) in CTLs deficient in both let-7 and Eomes generated from P14^+^CD4CRE^+^Eomes^fl/fl^Lin28Tg mice and assessed the phenotype. We found that Eomes re-expression enhanced the cytotoxic response and rescued expression of effector molecules. These results are shown in a new Figure 6—figure supplement 2. Furthermore, we also generated and expressed mutEomes (with silence mutation that disrupts the let-7 binding motif in the ORF of Eomes) in Let-7 transgenic CTLs from P14^+^Let-7Tg mice. Surprisingly, we observed only partial increase in the cytotoxic function of let-7Tg CTLs suggesting that let-7 miRNAs control CTL function through Eomes dependent and independent mechanisms. These data are shown in a new Figure 6—figure supplement 2.

6) The authors should be careful about making conclusions about quiescence of CD8 cells based on Ki67 alone and other markers (such as CD44 and CD122). Why would CD44 expression be significantly changed in spleen but not lymph node (Figure 1—figure supplement 2)? A direct assay of cell quiescence could be used. A direct assay of cell quiescence should be included in the analysis.

This is a great point. To directly assess the difference in cell cycle progression of naive T cells with and without let-7, we measured BrdU incorporation in vivo and included these results in a new Figure 1. Specifically, let-7 deficient cells had a higher frequency of BrdU^+^ CD8 T cells in comparison to the WT control. We also improved the statistics in Figure 1—figure supplement 3 by analyzing more samples, please see a new Figure 1—figure supplement 3. Thus, our data demonstrate that let-7 deficiency causes elevated expression of the activation markers, CD44 and CD122 and a significant increase of Ki67+ and BrdU^+^ cells in the population of naïve CD8 lymphocytes, suggesting that indeed let-7 controls the quiescent state of naïve CD8 T cells.

7) Many metabolic genes are differentially expressed in let-7 and lin28 Tg CD8 T cells. Do the authors have any metabolic analyses on a cellular level (such as OCR and ECR measurements with Seahorse) for WT, let-7 and lin28 Tg CD8 T cells that would help substantiate let-7 as a regulator of the metabolic reprograming that occurs in activated CD8 T cells?

Although we completely agree with the reviewers and understand that these experiments will further improve our manuscript, we are not be able to measure OCR or ECAR in our cells since we have no access to a Seahorse instrument. We also think that the observed let-7 mediated changes in metabolic targets of Myc give us sufficient evidence to at least suggest that let-7 may influence CD8 T cell metabolism through Myc. In order to answer this comment, we changed the wording in the manuscript to emphasize that our data is more speculative/suggestive of a let-7 mediated mechanism of metabolic reprogramming.

[Editors' note: further revisions were requested prior to acceptance, as described below.]

All reviewers felt that the manuscript was improved by the revision. The data within this manuscript are sufficient to support your model wherein T cell receptor-mediated downregulation of let-7 is important to facilitate CD8 T cell clonal expansion and the acquisition of effector function. Moreover, your discovery that this is at least in part due to derepression of the newly identified let-7 target, Eomes, is an important finding.

There are still some minor issues that need to be dealt with prior to publication, but its anticipated that these can be addressed without additional experiments.

You nicely show that mutated Eomes (with a disrupted let-7 binding motif) can increase the cytotoxic function of let-7Tg CTLs. However, a side-by-side comparison using WT-Eomes may have been even more informative. Is this data available? If so this would strengthen conclusions.

We completely agree with the reviewers. Unfortunately, we do not have this additional control, the generation of which would require long term experimental procedures.

In response to point #1, you wrote: "We found that Let-7 re-expression dramatically suppresses the expression of Eomes and all tested effector molecules in Lin28Tg CTLs."

In the previous version of the manuscript the impact of let-7 on the expression of the critical molecule, perforin was tested only on mRNA levels in CTLs. We decided to take the initiative to include the WB analysis of perforin expression on protein levels (please see the new Figure 4).

However, there is no analysis of Myc expression in 4Tg mice.

We really appreciate the comment. Our analysis of the expression of effector molecules in 4Tg CTLs was intentionally limited to day 5. At this time point (day 5) Myc expression in effector T cells is already gone (Chou et al., 2013). Unfortunately, we do not have RNA samples from early time points of 4Tg CD8 T cell activation to address this comment directly. Conducting this supplementary experiment again with three repeats will require isolation of CD8 T cells from VERY rare 4Tg mice (P14TgLin28TgLet7TgM2rtTATgRAG2KO) in our colony. However, we think that together, published evidence (let-7 targets Myc mRNA) and our data shown in Figure 3(let-7Tg expression in activated T cells suppresses Myc expression/ function) are sufficient to conclude that let-7 impacts the expression of Myc in activated T cells. Please note that the presence of let-7Tg T cells as an appropriate control that opposes the phenotype of Lin28Tg cells in these experiments leaves little room to doubt our conclusions.

Furthermore, Rao published that Let-7 targets Dicer mRNA and thus impairs global miRNA processing, which also promotes the CTL program. Do the authors have any data on Myc or Dicer? Do you agree that Dicer is an important target in this regard? See Trifari et al., 2013.

We thank the reviewers for the interesting point. However, we found that TCR- mediated activation suppresses the expression of all members of let-7 family during T cell differentiation into CTLs. Although we did not analyze Dicer expression, it seems very unlikely that Dicer goes down in activated (let-7 low) T cells due to the let-7 interference. In fact, we think that the enhanced CTL program in dicer deficient cells is due to the absence of let-7 miRNAs (among many others), which further emphasizes our conclusion. We included the last point in our Discussion.

There is still some imprecise language in the paper. You use the term "loss of quiescence" which is not accurate here. BrDU, like Ki67, indicates proliferation; authors should thus comment only on proliferation rather than "loss of quiescence" because the latter suggests a molecular mechanism. For example, it has been reported that Foxp1 is required for enforcing quiescence on naive CD8 T cells, but you don't present any data about that.

We agree, we do not know the molecular mechanisms of let-7 mediated impact on the naïve state of CD8 T cells. In fact, this is an exciting future area of investigation for our lab. It is known that naive T cells are in a quiescent state based on cell size, proliferation rate and activation markers. In fact, a similar phenotype was seen in Foxp1 deficient CD8 T cells. We found that Lin28Tg antigen-inexperienced CD8 T cells from LNs have:

– increased proliferation rate

– upregulated markers of activation

– increased cell size

All these results point towards the let-7 mediated molecular mechanism at play that enforces quiescence in naïve CD8 T cells. Taking under consideration reviewers comments we changed the text and used a more suggestive tone. However, we would still like to use term “loss of quiescence” in the Discussion section of our manuscript.

Similarly, you should not write "exit from the G0 stage of the cell cycle" since they did not have any data about G0 stage of the cell cycle. Furthermore, you like to use the term "let-7 deficiency." You should not encourage readers to believe that Lin28Tg is equivalent to let-7 deficiency because as discussed already Lin28 most likely affects not only let-7.

We agree and changed the text accordingly.

*In the Abstract, the phase "let-7 microRNAs as a novel molecular transmitter of TCR signaling" might be read as the opposite of the phenotype. As indicated in Figure 6—figure supplement 3, let-7 is drawn as a repressor. Thus, rather than a "transmitter", let-7 stops the signal / acts like a resistor in the circuit. The authors may want to find a term that better reflects the direction of the effect.*

Agreed, excellent point. We changed the Abstract.